# Physics-informed Temporal Alignment for Auto-regressive PDE Foundation Models

Congcong Zhu [*1 2 3]   Xiaoyan Xu [*1]   Jiayue Han [4]   Jingrun Chen [5 2 3]

## Abstract

Auto-regressive partial differential equation (PDE) foundation models have shown great potential in handling time-dependent data. However, these models suffer from error accumulation caused by the shortcut problem deeply rooted in auto-regressive prediction. The challenge becomes particularly evident for out-of-distribution data, as the pretraining performance may approach random model initialization for downstream tasks with long-term dynamics. To deal with this problem, we propose physics-informed temporal alignment (PITA), a self-supervised learning framework inspired by inverse problem solving. Specifically, PITA aligns the physical dynamics discovered at different time steps on each given PDE trajectory by integrating physics-informed constraints into the self-supervision signal. The alignment is derived from observation data without relying on known physics priors, indicating strong generalization ability to out-of-distribution data. Extensive experiments show that PITA significantly enhances the accuracy and robustness of existing foundation models on diverse time-dependent PDE data. The code is available at https://github.com/SCAILab-USTC/PITA.

## 1. Introduction

With the ongoing advancement in computational capabilities and data-driven methodologies (Cicirello, 2024; Brunton

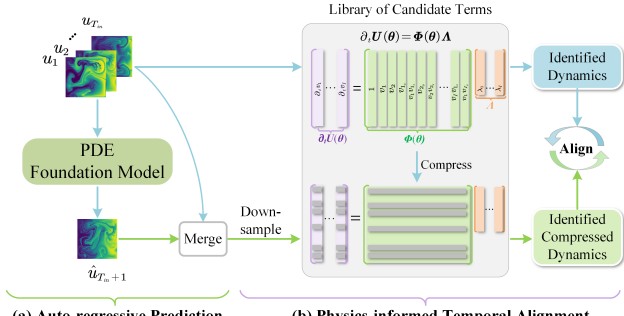

Figure 1: Insight of the proposed framework. Existing (a) Auto-regressive prediction may cause error accumulation for long-term PDE data. (b) Physics-informed temporal alignment is integrated to handle this problem.

& Kutz, 2024), PDE foundation models (Shen et al., 2024; Song et al., 2024; Gupta & Brandstetter, 2022) have gained increasing prominence in scientific machine learning. These foundational models leverage neural operators as surrogate models to perform PDE trajectory prediction, which runs significantly faster than the traditional numerical solvers. Among the various PDE foundation models, auto-regressive prediction, *i.e.*, predicting the next behavior based on past dynamics data, is one of the most promising strategies for training or pretraining, which aims to endow the models with generalization ability to downstream tasks (Geneva & Zabaras, 2020). However, it is observed that auto-regressive prediction may introduce a deeply-rooted shortcut problem, where the model chooses a simple and mendacious solution to approximately satisfy the optimization objective by duplicating previous dynamics. More details are discussed in Appendix B. Since the duplicating operation is easy for neural networks to learn, the model may overlook capturing long-term dynamics, resulting in error accumulation. Therefore, the long-term accurate prediction of auto-regressive PDE foundation models remains a significant challenge when facing downstream tasks with long trajectory data.

As we know, all trajectories of PDE data are simulated based on the underlying physical dynamics. Thus, the unique underlying physical equation can govern any segment in the corresponding PDE trajectory. This property naturally facil-

---

[*]Equal contribution [1]School of Artificial Intelligence and Data Science, University of Science and Technology of China [2]Suzhou Institute for Advanced Research, Unive rsity of Science and Technology of China [3]Suzhou Big Data & AI Research and Engineering Center [4]Department of Radiation Oncology, University of Kansas Medical Center [5]School of Mathematical Sciences, University of Science and Technology of China. Correspondence to: Jingrun Chen <jingrunchen@ustc.edu.cn>.

*Proceedings of the 42nd International Conference on Machine Learning*, Vancouver, Canada. PMLR 267, 2025. Copyright 2025 by the author(s).

itates embedding physical constraints into the PDE trajectory prediction. Early research such as (Raissi et al., 2019; Cuomo et al., 2022) utilizes known physics information to construct optimization objectives, enabling precise solutions for specific PDE trajectories. However, this approach may overfit the training data with known physics laws, preventing the models from serving as a foundation for downstream tasks. To this end, some foundation models (Song et al., 2024; Cao et al., 2024; Zhou et al., 2024; Sun et al., 2024) introduce PDE symbols as a condition prior to encoding into embedded representations by neural networks, thus guiding the prediction results. Although these methods flexibly encode various known PDE symbols to impose physical constraints on the prediction results, they merely perform conditional mapping based on known equation priors rather than capturing the intrinsic dynamics that represent PDE trajectory data. They may encounter significant performance degradation for downstream tasks without known physical priors. Therefore, exploring universal physical constraints is critically important for PDE foundation models.

Inspired by solving inverse problem (Rudy et al., 2017; Stephany & Earls, 2022), we thought over how to derive unknown dynamics systems from observation data and transform them into physics-informed constraints during auto-regressive prediction. To achieve this, we propose a self-supervised learning framework named physics-informed temporal alignment (PITA), which integrates governing laws discovery into auto-regressive prediction, as shown in Figure 1. In detail, PITA first discovers the governing PDE equation given the initial values with an unknown dynamics system by time series measurements. This process identifies the key derivative terms and parameters that form the structure and explicit expression of the PDE. Then, the auto-regressive prediction results are merged into the initial value sequence and used to rediscover the governing equation. Since the initial value sequence and the merged sequence are represented by the same underlying physical law, any discrepancy between the discovered governing equations corresponding to them indicates that auto-regressive prediction results may contain dynamics deviation. Based on this insight, we utilize the discrepancies of physical laws discovered from observation data and auto-regressive predictions to supervise the auto-regressive PDE models. Moreover, the discovered governing equation can also serve as physics-informed regularization, added to the optimization objective to supervise the auto-regressive predictions to follow the physical laws described. Finally, leveraging uncertainty-based weighting and alternating direction optimization, the physical supervisions derived from observation data are integrated into data-driven auto-regressive prediction. This ensures that the predictions follow both the observed temporal dynamics and the underlying physics laws, making PITA applicable to diverse time-dependent PDE data. In this way, PITA seamlessly unifies data-driven forecasting with physics-based modeling, offering a versatile framework for reliable long-term dynamics prediction.

## 2. Related Work

### 2.1. PDE Foundation Models

PDE foundation models have shown immense potential in many fields, such as fluid dynamics (Liu et al., 2024b; Luo et al., 2023), geophysics (Liu & Ma, 2024), solid mechanics (Yizheng et al., 2024), and chemical reactor modelling (Wang & Wu, 2024). Recently, PDE foundation models have been proposed to enable operator learning across different PDE families, incorporating pretraining and finetuning to enhance generalization. For instance, PIMRL (Wan et al., 2025) introduces a two-stages multi-scale learning framework that leverages multi-scale data for spatiotemporal dynamics prediction. PROSE (Sun et al., 2024) applies an encoder-decoder framework to integrate numerical data and equation embeddings, while PROSE-FD (Liu et al., 2024b) extends this approach to develop a foundation model for fluid dynamics. Moreover, MPP (McCabe et al., 2023) adopts an auto-regressive approach for pretraining and finetuning on time-dependent PDE datasets. To further investigate the generalization of auto-regressive prediction, DPOT (Hao et al., 2024) introduces a denoising pretraining strategy to improve transferability for downstream tasks. Despite the aforementioned advancements, there are many other works (Yang et al., 2023; Subramanian et al., 2024; Song et al., 2024; Ye et al., 2024b;a; Chen et al., 2024; Cao et al., 2024; Shen et al., 2024; Zhao et al., 2023) contributing to the development of PDE foundation models across various domains.

### 2.2. Shortcut Problem and Error Accumulation

Shortcuts commonly emerge as decision rules allowing models to excel on standard benchmarks while struggling to generalize under more complex testing conditions. This discrepancy between intended and learned solutions (Geirhos et al., 2020) leads to severe error accumulation, particularly in long temporal sequences. Several frameworks have been proposed to address shortcut learning. For example, the LTGR framework (Du et al., 2021) prevents overconfident predictions on shortcut samples, while COMI (Zhao et al., 2024) reduces the model's reliance on shortcuts by integrating standard empirical risk minimization to enhance its ability to extract underlying information. Additionally, diffusion-based and auto-regressive generative classifiers have been proposed to address shortcut issues by modeling both causal and spurious features (Li et al., 2024). Other works (Luo et al., 2021; Brown et al., 2023; Du et al., 2021; Dagaev et al., 2023; Robinson et al., 2021; Hermann et al., 2023; Scimeca et al., 2023; Chuah et al., 2022) have also

explored various strategies to detect and rectify shortcut learning.

## 2.3. Data-driven Inverse Problems

Modeling complex dynamical systems has traditionally relied on PDEs. Recently, data-driven techniques have emerged to uncover the underlying PDEs efficiently (Long et al., 2018; Brunton & Kutz, 2024). Sparse regression methods have leveraged finite differences to approximate partial derivatives and employed sparsity-promoting algorithms such as the Douglas–Rachford algorithm (Schaeffer, 2017) and Sparse Threshold Ridge Regression (STRidge) (Rudy et al., 2017) to identify PDE coefficients.

Physics-informed neural network (PINN) (Raissi et al., 2019) has been integrated into PDE discovery to address these limitations. For instance, DeepMoD (Both et al., 2021) combines PINNs and SINDy by introducing regularization terms into the PINN loss function. PINN-SR (Chen et al., 2021) adopts a similar loss function structure and introduces an alternating direction optimization training strategy.

# 3. Methodology

## 3.1. Overview

The general form of PDEs is considered as follows:

$$
\begin{aligned}
\frac{\partial \boldsymbol{u}(\boldsymbol{x}, t)}{\partial t} &= \mathcal{F}[\boldsymbol{u}](\boldsymbol{x}, t), \\
\boldsymbol{u}(\boldsymbol{x}, 0) &= \boldsymbol{u}_0(\boldsymbol{x}), \qquad \boldsymbol{x} \in \Omega, \\
\mathcal{B}[\boldsymbol{u}](\boldsymbol{x}, t) &= 0, \qquad \boldsymbol{x} \in \partial\Omega,
\end{aligned}
\tag{1}
$$

where $\boldsymbol{x} \in \Omega \subset \mathbb{R}^d$ denotes the spatial variable, $\boldsymbol{u} : [0, T] \times \Omega \to \mathbb{R}^{d_u}$ is the solution of a time-dependent PDE. $\mathcal{F}[\boldsymbol{u}](\boldsymbol{x}, t) = F(t, \boldsymbol{x}, \boldsymbol{u}, \partial_{\boldsymbol{x}}\boldsymbol{u}, \partial_{\boldsymbol{xx}}\boldsymbol{u}, \cdots)$ is a differential operator with spatial derivative terms. The initial condition is given by $\boldsymbol{u}_0(\boldsymbol{x}) : \Omega \to \mathbb{R}^{d_u}$, and the boundary condition is defined by the operator $\mathcal{B}$.

We propose PITA, a physics-informed temporal alignment strategy for PDE foundation models. PITA integrates auto-regressive prediction and PDE discovery into a self-supervised learning framework, as shown in Figure 2. This physics-informed alignment ensures that the predictions conform to both the observed temporal dynamics and the underlying PDEs, thereby enhancing predictive accuracy and maintaining physical fidelity.

The auto-regressive prediction, detailed in Sec. 3.2, enables iterative future state predictions by progressively leveraging prior outputs. To mitigate error accumulation inherent in the auto-regressive approach, PDE discovery is incorporated as discussed in Sec. 3.3. This integration encompasses several key steps, including compressing data across the full spa-

tiotemporal domain, constructing tailored function libraries, and employing sparse regression to identify dominant dynamics. Finally, the loss functions comprising data loss, physics loss, and consistency loss are introduced in Sec. 3.4. Together, these losses guide the model to balance among learning from observed data, enforcing physical constraints, and ensuring temporal coherence.

## 3.2. Auto-regressive Prediction

In the auto-regressive prediction, a neural operator $\mathcal{G}_{\boldsymbol{\theta}}$, parameterized by weights $\boldsymbol{\theta}$, takes $T_{in}$ frames as input and predicts the next frame based on the previous frames:

$$
\hat{\boldsymbol{u}}_{t+1} = \mathcal{G}_{\boldsymbol{\theta}}\left(\boldsymbol{u}_{<t}\right),
$$

where $\boldsymbol{u}_{<t}$ represents $\{\boldsymbol{u}_i\}_{i=t-T_{in}+1}^{t}$. When $t = T_{in}$, the input sequence denotes the ground truth frames, ensuring that the model starts with accurate information. $\boldsymbol{\theta}$ denotes the parameters of the neural operator. To predict the $(t+2)$-th frame, the neural operator uses $T_{in}$ frames along with the predicted frame $\hat{\boldsymbol{u}}_{t+1}$ as input. By iteratively performing this step, the model rolls out a time window of length $T_{ar}$, which can be customized. In our experiments, the roll-in window length is set to $T_{in} = 10$, and the roll-out window length is set to $T_{ar} = 1$ or $T_{ar} = 10$. However, directly applying this strategy may lead to the shortcut problem, causing the accumulation of errors propagated across the time window. Our framework employs a physics-informed temporal alignment strategy to address this issue.

## 3.3. Governing Equations Discovery

Following the auto-regressive predictions, the process of PDE discovery is employed to identify the underlying physical laws from the data while ensuring temporal alignment. First, the data is downsampled in both the spatial and temporal domains to enhance computational efficiency. Next, a nonlinear library is constructed, incorporating candidate terms that may represent the underlying physical dynamics. Finally, sparse regression is applied to derive the fundamental form of the PDE. Note that downsampling is not employed for the initial input sequence, so as to ensure the accuracy of the PDE discovered from ground truth data.

**Downsampling Data:** Before obtaining the PDE coefficients, the data generated through the auto-regressive prediction is downsampled to reduce computational time. For each $\boldsymbol{u}_t$, we assume that the function $\boldsymbol{u}_t$ is discretized over $n$ points, represented as $X_n = \{\boldsymbol{x}_1, \boldsymbol{x}_2, \ldots, \boldsymbol{x}_n\}$, $\boldsymbol{x}_i \in \mathbb{R}^d$. In the spatial domain, the grid is randomly sampled to reduce the number of points to one-quarter of its original size. The sampled grid is denoted as $\tilde{X}_w = \{\boldsymbol{x}_{i_1}, \boldsymbol{x}_{i_2}, \cdots, \boldsymbol{x}_{i_w}\}$, where $w = \lceil n/4 \rceil$ and $\{i_1, \cdots, i_w\} \subseteq \{1, 2, \cdots, n\}$ represents the set of sampled indices. The selection of $w$ will be discussed in Sec. 4.4. The derivatives are computed

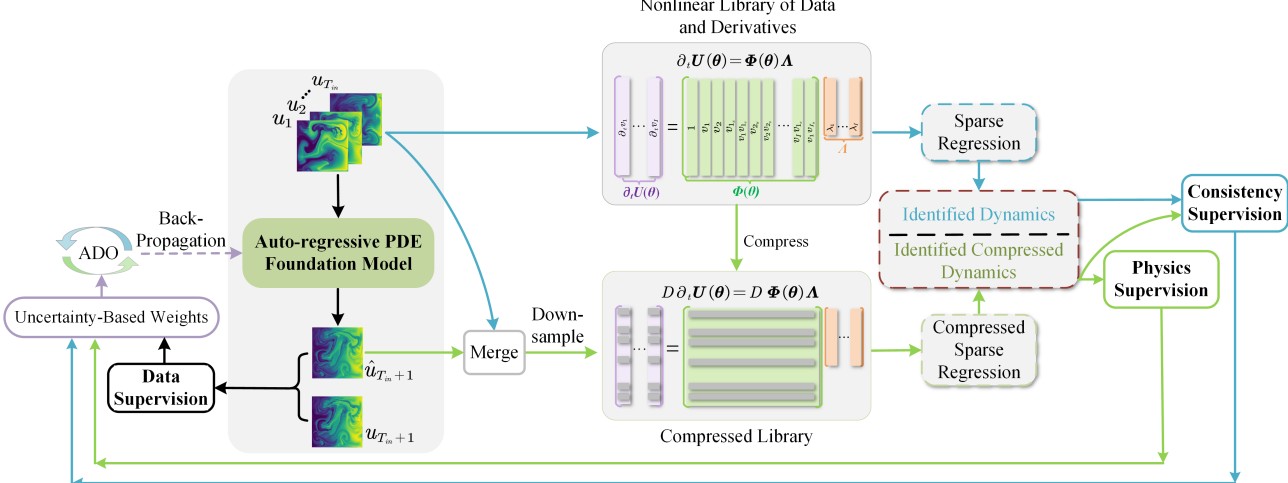

Figure 2: Work flow of PITA. The proposed framework integrates auto-regressive prediction and PDE discovery with self-supervised learning: (1) The pretrained PDE model takes the initial temporal states $\{u_t\}_{t=1}^{T_{in}}$ as input and predicts future states $\{\hat{u}_t\}_{t=T_{in}+1}^{T_{in}+T_{ar}}$ as output in an auto-regressive manner; (2) Data-driven PDE discovery is then performed on the compressed input sequence to infer the governing equations. Temporal alignment is achieved by matching the discovered physical laws from predictions with those obtained from the ground truth sequence; (3) The loss function consists of three parts, *i.e.*, data loss $\mathcal{L}_{Data}$, physics loss $\mathcal{L}_{Phy}$, and consistency loss $\mathcal{L}_{Con}$, with an uncertainty-based strategy employed to adjust the weights dynamically.

using a small number of spatially localized points near each measurement position via polynomial interpolation (Rudy et al., 2017). In the temporal domain, the most recent $T_C$ frames are retained, enabling the capture of critical temporal information that is more relevant than the entire sequence. As shown in Figure 2, the diagonal matrix $D$ signifies the downsampling process, with $D_{i,j} \in \{0,1\}$ for $i = j$ and $D_{ij} = 0$ for $i \neq j$, where non-zero diagonal elements indicate retaining data. For simplicity, the matrix $D$ is omitted in the following contexts.

**Building Libraries of Candidate Terms:** Before constructing libraries, a preparatory step involves rearranging all the compressed data $\tilde{U}(\theta) \in \mathbb{R}^{n \times m \times C}$ into a matrix $U(\theta) \in \mathbb{R}^{(n \times m) \times C}$, which represents $C$ physical variables collected over $n$ spatial locations and $m$ time points.

Next, we construct a library $\Phi(\theta) \in \mathbb{R}^{(n \times m) \times S}$ consisting of $S$ predefined candidate linear and nonlinear terms, along with partial derivatives for the PDE (Rudy et al., 2017). For instance, $\Phi(\theta)$ may include time derivatives, spatial derivatives, and $n$-th degree polynomial terms (Rudy et al., 2017), assembled in a matrix represented by

$$\Phi(\theta) = \left[ U(\theta), \dots, \partial_x U(\theta), U(\theta)\partial_x U(\theta), \dots, \mathbf{1} \right], \quad (2)$$

where finite differences are used to compute the derivatives. Each column of $\Phi(\theta)$ contains all of the values of a particular candidate function across all $(n \times m)$ space-time grid points where the data are collected.

**Sparse Regression:** After constructing the libraries for the PDE, the equation can be expressed as

$$\partial_t U(\theta) = \Phi(\theta)\Lambda, \quad (3)$$

where $\Phi(\theta)$ is the library of candidate terms, and $\Lambda \in \mathbb{R}^{S \times C}$ represents the sparse coefficient matrix. Assuming that only a few key terms dominate the underlying dynamics, the discovery problem is formulated as a sparse regression to identify the sparse coefficient matrix $\Lambda$:

$$\Lambda = \arg\min_{\Lambda} \|\Phi(\theta)\Lambda - \partial_t U(\theta)\|_2^2 + \alpha \|\Lambda\|_0. \quad (4)$$

This optimization problem is addressed using sparse regression (Rudy et al., 2017; Ma et al., 2023; Zhang, 2008). For example, if $C = I$ where $U = [U_1, \dots, U_I]$, $\Lambda = [\lambda_1, \dots, \lambda_I]$, where $U_i \in \mathbb{R}^{(n \times m) \times 1}$, $\lambda_i \in \mathbb{R}^{S \times 1}$. Equation (3) can be rewritten as

$$[\partial_t U_1(\theta), \dots, \partial_t U_I(\theta)] = \Phi(\theta) [\lambda_1, \dots, \lambda_I]. \quad (5)$$

Thus the coefficient matrix $\Lambda$ is solved iteratively through each $\lambda_i$, as detailed in Algorithm 1. This approach integrates ridge regression with hard threshold to enforce sparsity, striking a balance between model complexity and accuracy.

### 3.4. Integrating Data and Physics Supervisions

The total loss function integrates three parts: the data loss $\mathcal{L}_{Data}$, the physics loss $\mathcal{L}_{Phy}$ and the consistency loss $\mathcal{L}_{Con}$.

**Algorithm 1** Sparse regression for Equation (4) and (5).

---

**Input:** Time derivative vector $\partial_t \boldsymbol{U}_i(\boldsymbol{\theta})$, candidate function library matrix $\boldsymbol{\Phi}(\boldsymbol{\theta})$, threshold tolerance $\beta$, maximum iteration number $K$.

**Initialize:** $\boldsymbol{\lambda}_i = \boldsymbol{\Phi}^\dagger(\boldsymbol{\theta})\partial_t \boldsymbol{U}_i(\boldsymbol{\theta}), i = 1, \cdots, I, k = 0$.

**for** $k \leqslant K$ **do**

    Determine two groups of indices of coefficients in $\boldsymbol{\lambda}_i$:
    $\mathcal{P} = \{p \in \mathcal{P} : |\lambda_i^p| < \beta\}, \mathcal{Q} = \{q \in \mathcal{Q} : |\lambda_i^q| \geqslant \beta\}$.

    Impose sparsity on small values:
    $\boldsymbol{\lambda}_i^{\mathcal{P}} = \boldsymbol{0}$.

    Update $\boldsymbol{\lambda}_i^{\mathcal{Q}}$:
    $\boldsymbol{\lambda}_i^{\mathcal{Q}} = \arg\min_{\boldsymbol{\lambda}_i^{\mathcal{Q}}} \{\|\boldsymbol{\Phi}^{\mathcal{Q}}(\boldsymbol{\theta})\boldsymbol{\lambda}_i^{\mathcal{Q}} - \partial_t \boldsymbol{U}_i^{\mathcal{Q}}(\boldsymbol{\theta})\|_2^2 + \alpha\|\boldsymbol{\lambda}_i^{\mathcal{Q}}\|_0\}$.

    Update $k = k + 1$.

**end for**

**Output:** The best solution $\boldsymbol{\lambda}_i = \boldsymbol{\lambda}_i^{\mathcal{P}} \cup \boldsymbol{\lambda}_i^{\mathcal{Q}}$.

---

**Data Loss**: The data loss is defined as

$$\mathcal{L}_{Data}(\boldsymbol{\theta}) = \frac{1}{T_{ar}} \sum_{i=1}^{T_{ar}} \frac{\|\boldsymbol{u}_{i+T_{in}} - \hat{\boldsymbol{u}}_{i+T_{in}}(\boldsymbol{\theta})\|_2}{\|\boldsymbol{u}_{i+T_{in}}\|_2}, \quad (6)$$

where $\hat{\boldsymbol{u}}_{i+T_{in}}(\boldsymbol{\theta})$ represents the predicted solution and $\boldsymbol{u}_{i+T_{in}}$ is the ground truth data. This term quantifies the discrepancy between predictions and observations. Previous works have employed similar approaches such as (Raissi et al., 2019; Chen et al., 2021; Both et al., 2021).

**Physics Loss:** Inspired by (Rudy et al., 2017; Brunton et al., 2016; Chen et al., 2021; Raissi et al., 2019), the physics loss is expressed as

$$\mathcal{L}_{Phy}(\boldsymbol{\theta}) = \sum_{i=1}^{T_{ar}} \|\partial_t \boldsymbol{U}_i(\boldsymbol{\theta}) - \boldsymbol{\Phi}_i(\boldsymbol{\theta})\boldsymbol{\Lambda}_i\|_2^2 + \alpha\|\boldsymbol{\Lambda}_i\|_0, \quad (7)$$

where the residual term ensures that the product of the computed coefficients $\boldsymbol{\Lambda}_i$ and the function library $\boldsymbol{\Phi}_i(\boldsymbol{\theta})$ approximates the time-dependent term $\partial_t \boldsymbol{U}_i(\boldsymbol{\theta})$. The $\ell_0$ norm enforces sparsity in the coefficients.

**Consistency Loss:** The consistency loss is defined as

$$\mathcal{L}_{Con}(\boldsymbol{\theta}) = \sum_{i=1}^{T_{ar}} \|\boldsymbol{\Lambda}^* - \boldsymbol{\Lambda}_i(\boldsymbol{\theta})\|_2^2, \quad (8)$$

where $\boldsymbol{\Lambda}^*$ denotes the true coefficients from ground-truth data and $\boldsymbol{\Lambda}_i(\boldsymbol{\theta})$ are from the $i$-th time window, both obtained via sparse regression as detailed in Algorithm 1. This loss enforces consistency between the discovered physics laws represented by $\boldsymbol{\Lambda}_i(\boldsymbol{\theta})$ and real physics laws represented by $\boldsymbol{\Lambda}^*$. To the best of our knowledge, it is the first work to use time alignment as a constraint for PDE foundation models.

**Uncertainty-Based Weighted Loss Function**: The total loss function combines these three supervisions to balance the different learning tasks of the model. Determining the penalty coefficients for each term is challenging, as model performance across tasks strongly depends on the relative weighting of each loss (Sener & Koltun, 2018; Chen et al., 2018). Instead of relying on traditional hyperparameter tuning, we employ an uncertainty-based multi-task learning strategy (Kendall et al., 2018; Liebel & Körner, 2018) to adjust these weights dynamically.

Let $\boldsymbol{\delta} = \{\delta_1, \delta_2, \delta_3\}$ be the parameters for the weights of the three losses in our task. The weighted total loss function becomes

$$\mathcal{L}_{Total}(\boldsymbol{\theta}, \boldsymbol{\delta}) = \frac{1}{2\delta_1^2}\mathcal{L}_{Data}(\boldsymbol{\theta}) + \frac{1}{2\delta_2^2}\mathcal{L}_{Phy}(\boldsymbol{\theta}) + \frac{1}{2\delta_3^2}\mathcal{L}_{Con}(\boldsymbol{\theta}) + \log\delta_1\delta_2\delta_3. \quad (9)$$

These parameters are updated simultaneously with the model parameters during training. Minimizing this objective with respect to $\delta_1$, $\delta_2$, and $\delta_3$ allows the model to adaptively learn the relative weights of the three losses based on the data, which enables robustness across varying datasets.

### 3.5. Alternating Direction Optimization

The total loss function described in Equation (9) exhibits an implicit and complex form, which complicates the direct resolution of the optimization problem due to the $\ell_0$ regularization, rendering it NP-hard (Chen et al., 2021). Although relaxing the $\ell_0$ term through the less stringent $\ell_1$ regularization enhances well-posedness and allows for optimization in a continuous space, this may lead to false-positive identification, hindering the accurate realization of the sparsity of the PDE coefficients (Berg & Nyström, 2019; Both et al., 2021). Inspired by (Chen et al., 2021), we employ an alternating direction optimization (ADO) algorithm to decompose the overall optimization problem into a series of manageable subproblems, enabling the sequential optimization of $\boldsymbol{\theta}$ and $\boldsymbol{\Lambda}$ over several alternating iterations (denoted as $k$). For instance, in the $(k+1)$-th alternating iteration, the sparse coefficient matrix $\boldsymbol{\Lambda}$ in Equation (11) is updated to $\boldsymbol{\Lambda}_{k+1}$ using sparse regression, based on the parameters $\boldsymbol{\theta}_k$ of the neural operator $\mathcal{G}_{\boldsymbol{\theta}}$ from the previous iteration:

$$\boldsymbol{\Lambda}_{k+1} := \arg\min_{\boldsymbol{\Lambda}}[\|\boldsymbol{\Phi}(\boldsymbol{\theta}_k)\boldsymbol{\Lambda} - \partial_t \boldsymbol{U}(\boldsymbol{\theta}_k)\|_2^2 + \alpha\|\boldsymbol{\Lambda}\|_0]. \quad (10)$$

Next, the parameters $\boldsymbol{\theta}$ in the current iteration are then updated to $\boldsymbol{\theta}_{k+1}$ based on the solved $\boldsymbol{\Lambda}_{k+1}$,

$$\begin{aligned} \boldsymbol{\theta}_{k+1}, \boldsymbol{\delta}_{k+1} := \arg\min_{\boldsymbol{\theta}, \boldsymbol{\delta}}[&\frac{1}{2\delta_1^2}\mathcal{L}_{Data}(\boldsymbol{\theta}, \boldsymbol{\Lambda}_{k+1}) \\ &+ \frac{1}{2\delta_2^2}\mathcal{L}_{Phy}(\boldsymbol{\theta}, \boldsymbol{\Lambda}_{k+1}) \\ &+ \frac{1}{2\delta_3^2}\mathcal{L}_{Con}(\boldsymbol{\theta}, \boldsymbol{\Lambda}_{k+1}) + \log\delta_1\delta_2\delta_3]. \end{aligned} \quad (11)$$

Table 1: Comparison of PITA and existing auto-regressive strategies with $T_{ar} = 1$. The underlined results represent the predictions of PITA, while the bold results indicate the optimal outcomes for the same model. '-' means that the result is unavailable. Notably, five datasets exhibit long trajectory characteristics, with detailed descriptions provided in Appendix C.

| Model | Finetune Strategy | FNO-NS-$\nu$ | | | PDEBench | | PDEBench-CNS-$(M, \eta)$ | | | | | | PDEArena | | CFDBench |
|---|---|---|---|---|---|---|---|---|---|---|---|---|---|---|---|
| | | 1e-5 | 1e-4 | 1e-3 | DR | SWE | 1,0.1 | 1,0.01 | M1 | 0.1,0.1 | 0.1,0.01 | M0.1 | NS-Force | NS | - |
| *Predict short trajectory* | | | | | | | | | | | | | | | |
| DPOT-Ti 7M | Auto-regress | **0.05200** | **0.00385** | 0.00380 | **0.01326** | 0.00208 | **0.01120** | 0.01950 | **0.01535** | 0.01740 | 0.01380 | 0.01560 | **0.10269** | 0.09100 | 0.00391 |
| | PITA | 0.05629 | 0.00499 | 0.00218 | 0.02119 | **0.00202** | 0.01793 | **0.01565** | 0.01679 | **0.01080** | **0.01250** | **0.01165** | 0.11988 | **0.06465** | **0.00360** |
| DPOT-S 30M | Auto-regress | 0.03220 | 0.00641 | 0.00301 | **0.01239** | 0.00210 | 0.01290 | 0.01670 | 0.01480 | **0.0152** | 0.01260 | **0.01390** | 0.07598 | 0.08670 | 0.00382 |
| | PITA | **0.02240** | **0.00232** | **0.00157** | 0.01346 | **0.00150** | **0.01057** | **0.01217** | **0.01137** | 0.01750 | **0.01140** | 0.01445 | **0.06769** | **0.04190** | **0.00335** |
| MPP-Ti 7M | Auto-regress | - | - | - | 0.03510 | 0.00645 | - | - | 0.05841 | - | - | 0.04611 | - | - | - |
| | PITA | - | - | - | **0.03384** | **0.00605** | - | - | **0.04370** | - | - | **0.04025** | - | - | - |
| MPP-S 30M | Auto-regress | - | - | - | **0.02974** | 0.00281 | - | - | 0.04287 | - | - | 0.02012 | - | - | - |
| | PITA | - | - | - | 0.03024 | 0.00274 | - | - | **0.04051** | - | - | **0.01906** | - | - | - |
| DPOT-M 122M | Auto-regress | **0.02290** | 0.00385 | 0.00297 | 0.01209 | 0.00219 | **0.00998** | 0.01460 | 0.01230 | 0.01610 | **0.00947** | 0.01280 | **0.05571** | 0.02940 | 0.00373 |
| | PITA | 0.03199 | **0.00185** | **0.00193** | **0.00957** | **0.00154** | 0.01101 | **0.01032** | **0.01067** | **0.00945** | 0.01010 | **0.00958** | 0.05829 | **0.02191** | **0.00289** |
| FNO-M 170M | Auto-regress | 0.08036 | 0.00577 | 0.00302 | 0.08608 | 0.00501 | 0.27849 | 0.03345 | 0.15597 | 0.13200 | 0.04449 | 0.08825 | **0.15190** | 0.15590 | 0.01382 |
| | PITA | **0.07668** | **0.00552** | **0.00152** | **0.07014** | **0.00423** | **0.10947** | 0.03483 | **0.07215** | **0.10224** | 0.05686 | **0.07955** | 0.15594 | **0.14066** | **0.00756** |
| DPOT-L 500M | Auto-regress | 0.02130 | 0.00400 | 0.00299 | **0.00801** | 0.00184 | 0.01080 | 0.01310 | 0.01195 | 0.01600 | **0.00905** | 0.01253 | 0.05298 | 0.02780 | 0.00322 |
| | PITA | **0.01059** | **0.00198** | **0.00139** | 0.01084 | 0.00172 | **0.01001** | **0.00995** | **0.00998** | 0.02004 | 0.01871 | 0.01938 | **0.04965** | **0.02048** | **0.00302** |
| *Predict long trajectory* | | | | | | | | | | | | | | | |
| DPOT-Ti 7M | Auto-regress | - | 0.03670 | 0.00580 | 0.01480 | 0.00241 | - | - | - | - | - | - | **0.30034** | - | - |
| | PITA | - | **0.01718** | **0.00327** | **0.01090** | **0.00199** | - | - | - | - | - | - | 0.31012 | - | - |
| DPOT-S 30M | Auto-regress | - | 0.02370 | 0.00437 | 0.01350 | 0.00235 | - | - | - | - | - | - | 0.26800 | - | - |
| | PITA | - | **0.00854** | **0.00223** | **0.00994** | **0.00137** | - | - | - | - | - | - | **0.22620** | - | - |
| MPP-Ti 7M | Auto-regress | - | - | - | 0.05212 | 0.03291 | - | - | - | - | - | - | - | - | - |
| | PITA | - | - | - | **0.03844** | **0.02174** | - | - | - | - | - | - | - | - | - |
| MPP-S 30M | Auto-regress | - | - | - | 0.04258 | 0.01631 | - | - | - | - | - | - | - | - | - |
| | PITA | - | - | - | **0.03704** | **0.00950** | - | - | - | - | - | - | - | - | - |
| DPOT-M 122M | Auto-regress | - | 0.0126 | 0.00335 | 0.01030 | 0.00227 | - | - | - | - | - | - | 0.17200 | - | - |
| | PITA | - | **0.00694** | - | **0.00904** | **0.00135** | - | - | - | - | - | - | **0.16390** | - | - |
| FNO-M 170M | Auto-regress | - | 0.01761 | 0.00425 | 0.04101 | 0.00924 | - | - | - | - | - | - | 0.43136 | - | - |
| | PITA | - | **0.01710** | **0.00227** | **0.02884** | **0.00614** | - | - | - | - | - | - | **0.38731** | - | - |
| DPOT-L 500M | Auto-regress | - | 0.0104 | 0.00323 | 0.00739 | 0.00170 | - | - | - | - | - | - | 0.17000 | - | - |
| | PITA | - | **0.00708** | **0.00199** | **0.00670** | **0.00121** | - | - | - | - | - | - | **0.16488** | - | - |

This alternation between the suboptimal solutions will converge towards a high-quality optimization result that satisfies global convergence.

## 4. Experiments

**Datasets:** To enable comparisons with baselines employing the auto-regressive strategy, we select 12 datasets from four different sources: 3 datasets from FNO (Li et al., 2020), 6 datasets from PDEBench (Takamoto et al., 2022), 2 datasets from PDEArena (Gupta & Brandstetter, 2022), and 1 dataset from CFDBench (Luo et al., 2023). For generalization tasks, the Burgers' equation from (Boussif et al., 2022) is included as an additional dataset. Additional details regarding the datasets used can be found in the Appendix C.

**Baselines**: We compare PITA with auto-regressive baselines, primarily focusing on PDE foundation models such as DPOT (Hao et al., 2024) and MPP (McCabe et al., 2023). For DPOT, we select models of sizes Ti, S, M, and L, which have been pretrained on 12 datasets. In the case of MPP, we choose the Ti and S models, pretrained on 10 datasets. Both models are designed to work predominantly with 2D datasets. Additionally, we assess our method on single-family models (Shen et al., 2024), where pretraining and finetuning are conducted on a single dataset instead of multiple datasets. FNO-M (Li et al., 2020) serves as the representative baseline. The sizes and parameters of the various models are detailed in Appendix D.1. These ensure a comprehensive and fair evaluation across diverse PDE settings.

**Training and Evaluation:** All experiments are carried out on a single A800 GPU with 80 GB of memory. We apply the commonly used scale-independent normalized root mean squared error (nRMSE) (Takamoto et al., 2023; Hao et al., 2023) to measure the quality of the prediction, which is defined as follows,

$$\text{nRMSE} = \frac{1}{T_{test}} \sum_{i=1}^{T_{test}} \frac{\|\boldsymbol{u}_i - \hat{\boldsymbol{u}}_i(\boldsymbol{\theta})\|_2}{\|\boldsymbol{u}_i\|_2}, \qquad (12)$$

where $T_{test}$ indicates the length of testing data in the temporal domain.

### 4.1. State-of-the-Art Results

The in-distribution performance of PITA is evaluated and presented in Table 1. The testing datasets comprise PDE data that share the same boundary conditions and parameters as the training data but differ in their initial conditions. Overall, PITA demonstrates state-of-the-art (SOTA) performance in both short trajectory predictions (10 steps) and long trajectory predictions (longer than 10 steps). A detailed analysis of the results is provided below.

**Short Trajectory Prediction:** When performing short trajectory prediction, PITA achieves SOTA performance across most datasets and various model sizes, particularly with medium and large models. For small-sized models (Ti, S), PITA achieves the best results in 75% of prediction tasks, with an average improvement of 11.21% in testing outcomes. Notably, the maximum improvement observed was an im-

pressive 63.81% with the DPOT-S model on the FNO-NS-1e-4 dataset. For medium and large-sized models (M, L), PITA excels in 77.78% of the prediction tasks, achieving an average reduction in nRMSE of 13.53%, with the best improvement recorded at 53.36% using the DPOT-L model on the FNO-NS-1e-3 dataset. These results indicate that although PITA is not specifically tailored for PDE data with relatively short trajectories, it still exhibits excellent performance compared to finetuning approaches that rely solely on the auto-regressive approach. This highlights PITA's robustness and effectiveness in handling a variety of prediction tasks, contributing to its overall superiority in performance.

**Long Trajectory Prediction:** By integrating temporal alignment with physics-informed constraints, PITA demonstrates superior performance over traditional auto-regressive training methods, effectively addressing the shortcut problem commonly encountered in long trajectory prediction.

Overall, PITA reduces the total accumulated error significantly compared to the auto-regressive approach, achieving an improvement of 30.22% over the long trajectory, which translates to an average decrease of 2.00% in error for each time step. The shortcut problem is particularly pronounced in the FNO-NS-1e-4 dataset, where the average percentage of error accumulation per prediction step reaches 34.31%. In contrast, the shortcut problem is less significant in the PDEBench-SWE and PDEBench-DR datasets that are longest trajectory datsets (91 steps). This discrepancy can be attributed to a main factor that the dynamics exhibits smaller variations in long term trajectory, making it easier for the model to capture the temporal dynamics (Takamoto et al., 2022). Despite this, PITA still demonstrates meaningful performance improvements over all the baselines. We further analyze the relevance between the performance gains of PITA and foundation model size. The experiments are conducted on PDEBench-DR with the longest trajectory, as shown in Figure 3. We can see that in long trajectory predictions, PITA demonstrates scalability. It even outperforms the baseline with 100M parameters when using only 30M parameters, showcasing parameter efficiency in modeling long-term dynamics. Additional scaling experiments on long trajectory datasets are presented in Appendix E.3.

### 4.2. Solution to Error Accumulations

Following the error visualization methodology introduced in (Christlieb et al., 2016), we compute and plot the rolling-step mean squared error (MSE) for each long-term forecasting dataset, as illustrated in Figure 10. This rolling window highlights temporal trends in prediction quality and makes error accumulation more apparent. Our analysis demonstrates that prediction errors on datasets such as PDEBench-SWE, FNO-NS-1e-3, and FNO-NS-1e-4 exhibit no significant error accumulation over time. For other

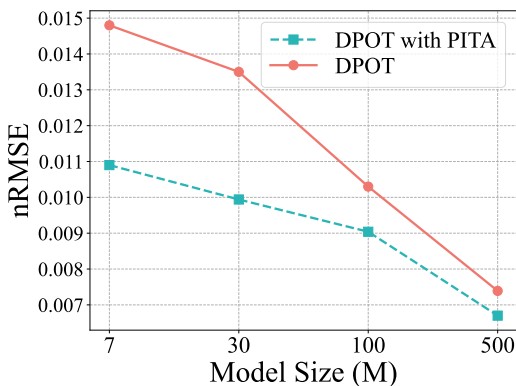

Figure 3: Results of scalability for different model sizes.

datasets, although errors display a gradual increase with temporal progression, PITA substantially reduces error accumulation compared to conventional auto-regressive benchmarks. These findings provide strong empirical evidence that PITA's multi-scale consistency mechanism effectively mitigates error propagation across extended temporal sequences in long-term forecasting tasks.

### 4.3. Generalizing to Downstream tasks

To investigate the generalization performance of PITA on downstream tasks, we consider two experimental settings: (1) PDEs of the same type as those in the pretraining datasets, but with different coefficients, (2) PDEs of a different type not present in the pretraining datasets.

In the first setting, we assess PITA's generalization capacity using the compressible Navier-Stokes equations from PDEBench (Takamoto et al., 2022) with different shear viscosity $\eta$ and bulk viscosity $\zeta$. The finetuning results, detailed in the fourth column of Table 2, demonstrate that PITA outperforms all other settings, achieving an average reduction in error by 5.93%. This exceptional performance highlights PITA's capability to proficiently adapt to the intricate challenges posed by compressible fluid dynamics.

In the second setting, we use the viscous Burgers' equation (Boussif et al., 2022) for evaluation. The pretrained model is finetuned on this dataset for 500 epochs employing both auto-regressive training and PITA. The corresponding results, presented in the third column of Table 2, demonstrate that PITA consistently outperforms across all model sizes, achieving an average error reduction of 37.8% compared to the original auto-regressive approach. The incorporation of PDE discovery enables PITA to effectively learn the underlying physical laws of the dataset without prior knowledge, indicating enhanced adaptability to various types of PDEs. These findings underscore PITA's robust adaptability and practical value for a broad range of physical modeling tasks.

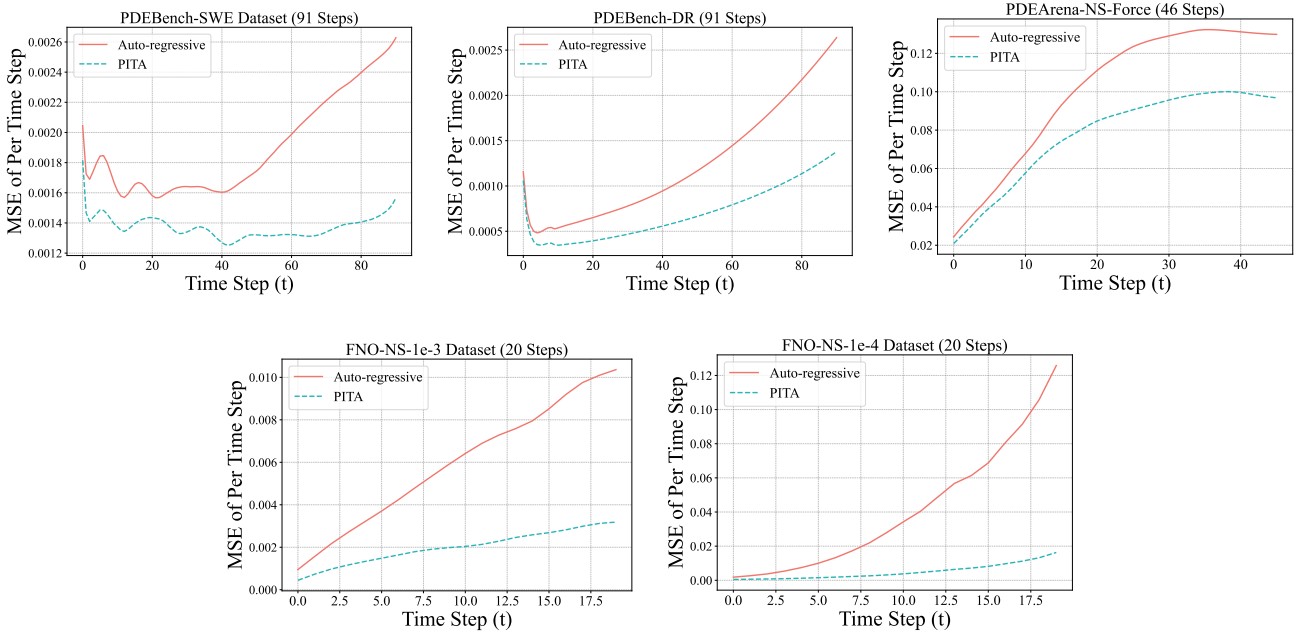

Figure 4: Rolling step MSE on long trajectory datasets.

Table 2: Generalization on two downstream tasks of PITA compared with baselines. More comprehensive generalization experiments are presented in Appendix E.4.

| Pretrained Model | Finetune Strategy | Burgers | PDE-Bench CNS-$(M,\eta)$ $1,1 \times 10^{-8}$ |
|---|---|---|---|
| DPOT-Ti | Auto-regress | 0.00904 | 0.18681 |
| | PITA | **0.00894** | **0.16498** |
| DPOT-S | Auto-regress | 0.00533 | 0.13372 |
| | PITA | **0.00458** | **0.12941** |
| DPOT-M | Auto-regress | 0.00378 | 0.14607 |
| | PITA | **0.00362** | **0.14453** |

## 4.4. Ablation Studies

We performed five ablation studies to assess the impact of different design choices in PITA, training small-scale models on the FNO-NS-1E-3 dataset for Tasks 1–4 and on the Burgers equation for Task 5.

**Task 1: Effectiveness of Loss Components**. To highlight the indispensability of each supervision, we systematically evaluate the pretrained model under various configurations. Specifically, $\mathcal{L}_{Data}$ refers to using only data loss, following the auto-regressive strategy. $\mathcal{L}_{Data}+\mathcal{L}_{Phy}$ denotes the combination of data loss and physics loss, while $\mathcal{L}_{Data} + \mathcal{L}_{Con}$ represents the combination of data loss and consistency loss. $\mathcal{L}_{Total}$ represents the inclusion of all three components, aligning with the PITA training strategy. Except for the $\mathcal{L}_{Data}$ setting, all configurations adopt the multi-task learning strategy described in Sec. 3.4. As shown in Table 3, PITA achieves the best results across all settings, underscoring the effectiveness of integrating multiple loss

components. Notably, the physics loss $\mathcal{L}_{Phy}$ plays a pivotal role in guiding the model to adhere to the underlying physical laws, resulting in 44.97% accuracy promotion. The consistency loss $\mathcal{L}_{Con}$ leads to a 44.10% increase in accuracy. It maintains temporal alignment and is essential for the stability of the learned solution, ensuring smooth transitions in long-term predictions.

**Task 2: Effects of Spatial Downsampling**. To boost computational efficiency, PITA adopts a downsampling strategy to randomly select grids in the spatial domain. The sampled grid is represented as $\tilde{X}_w = \{\boldsymbol{x}_{i_1}, \boldsymbol{x}_{i_2}, \ldots, \boldsymbol{x}_{i_w}\}$, where $w = \lceil n/l \rceil$ denotes the number of sampled points, and $l$ represents the downsampling factor. The value of $l$ is selected from $\{1, 2, 4\}$. Experimental results indicate that larger downsampling factors (i.e., fewer sampled grids) often lead to better performance, which can be attributed to the balance between data sparsity and generalization (Liu et al., 2024a). Fewer sampled grids encourage the model to focus on key spatial features, effectively mitigating overfitting to fine-grained noise present in denser grids.

**Task 3: Effects of Temporal Downsampling**. To enhance computational efficiency, PITA employs a downsampling strategy that selects frames within the temporal domain. The sampled solution is expressed as $\{\boldsymbol{u}_t\}_{t=T-T_C}^{T-1}$, where $T$ represents the last frame and $T_C$ denotes the number of frames retained during downsampling. The values of $T_C$ are selected from $\{10, 7, 5, 3\}$. Notably, the best results for both prediction tasks occur at $T_C = 7$. Conversely, increasing or decreasing $T_C$ from this optimal point results in elevated nRMSE values. While excessive downsampling can lead to

a significant loss of critical information, retaining too many frames may detract from the model's efficiency. Given that the results for $T_C = 3$ and $T_C = 7$ are relatively close, we opted to use $T_C = 3$ for the majority of our results presented in Table 1 to optimize computational costs.

**Task 4: Effects of Multi-Task Learning Strategy.** The influence of weighting different components of the total loss is evaluated. The data loss weight is set 1, the physics loss and consistency loss are weighted by $\alpha_1$ and $\alpha_2$, respectively. We compare manually chosen weights $(\alpha_1, \alpha_2)$ with the automatically adjusted strategy. The tested manual settings for $(\alpha_1, \alpha_2)$ are $(0.5, 0.5), (0.3, 0.7)$, and $(0.7, 0.3)$. As presented in Table 3, PITA achieves an average improvement of 9.06% over all manual weight settings, suggesting the effectiveness of its adaptive strategy in dynamically balancing competing objectives. This ensures each loss component contributes optimally to the learning process. Furthermore, PITA exhibits greater robustness to weight variations compared to existing methods (Raissi et al., 2019), highlighting its superior ability to generalize across diverse PDE data.

**Task 5: Effects of Key Library Terms.** We assessed how key library terms affects PITA's performance in inferring physical laws by conducting experiments with distinct configurations on the Burgers equation dataset, which was excluded from pre-training. The Burgers equation is composed of a time-dependent term $\partial_t \boldsymbol{u}$, a diffusion term $\beta\nabla\boldsymbol{u}$ and a convective term $\boldsymbol{u}\delta\boldsymbol{u}$, which are indispensable for capturing the underlying dynamics of the system (Gao & Zou, 2017). Table 3 compares three configurations: "None" removes all first- and second-order derivatives (no convective or diffusion terms), "One-Order" removes only second-order derivatives (diffusion eliminated, convection retained), and "Full" uses the complete library. It is evident that the full library achieves state-of-the-art predictive accuracy. In the "One-Order" configuration where only the convective term is retained, the performance experiences a slight degradation. Further removal of both the convective and diffusion terms leads to a modest compromise in the model's ability to capture the underlying dynamics. Nevertheless, the model still achieves a 41.42% improvement over the baseline, demonstrating the robustness of the approach even with limited incorporation of physical knowledge. These results concur with Task 1's ablation of physics loss, where PITA continues to improve over the baseline via data and consistency losses even when physics loss is ineffective.

### 4.5. Limitations

Despite the advantages mentioned above, the proposed PITA still has some limitations. First, PITA may require more computational cost, which results from an intricate gradient propagation process due to the additional two losses. Importantly, this extra computation time does not scale with model

Table 3: Results of the ablation studies. For each task, only the specified settings are different with the other hyperparameters and training configurations remaining the same.

| Task No. | Settings | nRMSE (10 Step) | nRMSE (Full Length) |
|---|---|---|---|
| TASK 1 | $\mathcal{L}_{Data}$ | 0.00301 | 0.00437 |
| | $\mathcal{L}_{Phy} + \mathcal{L}_{Data}$ | 0.00166 | 0.00240 |
| | $\mathcal{L}_{Con} + \mathcal{L}_{Data}$ | 0.00165 | 0.00249 |
| | $\mathcal{L}_{Total}$ | **0.00157** | **0.00223** |
| TASK 2 | $l = 1$ | 0.00169 | 0.00238 |
| | $l = 2$ | 0.00168 | 0.00237 |
| | $l = 4$ | **0.00157** | **0.00223** |
| TASK 3 | $T_C = 10$ | 0.00161 | 0.00229 |
| | $T_C = 7$ | **0.00155** | **0.00221** |
| | $T_C = 5$ | 0.00170 | 0.00240 |
| | $T_C = 3$ | 0.00157 | 0.00223 |
| TASK 4 | $(0.5, 0.5)$ | 0.00172 | 0.00243 |
| | $(0.3, 0.7)$ | 0.00175 | 0.00246 |
| | $(0.7, 0.3)$ | 0.00171 | 0.00242 |
| | Automatic | **0.00157** | **0.00223** |
| TASK 5 | None | 0.00325 | 0.01091 |
| | One-Order | 0.00287 | 0.00862 |
| | Complete | **0.00152** | **0.00776** |
| | Auto-regress | 0.00431 | 0.01857 |

size, indicating that PITA remains suitable for larger models. For example, PITA may take approximately 42.32% and 22.8% longer to process for DPOT-Ti and DPOT-L, respectively. Thus, an appropriate downsampling strategy in PDE discovery is advisable to balance effectiveness and computational consumption. Second, the performance of PITA may also be limited by the method used for PDE discovery. The completeness of the library of terms and the level of noise in the observational data may affect the model's performance.

## 5. Conclusion

In this paper, we present PITA, a novel approach for PDE foundation models. PITA is designed to address the limitations of traditional auto-regressive methods by mitigating shortcut issues. Instead of relying on prior knowledge embedded, PITA discovers underlying physical laws in a data-driven manner and incorporates these laws into optimization objectives, compelling the model to accurately learn the governing physics. Extensive experiments are conducted to rigorously validate PITA on foundation models pretrained across diverse datasets scaling up to 500 million parameters, as well as individual neural operators tailored to specific PDEs. PITA consistently achieves state-of-the-art performance across multiple datasets, demonstrating its effectiveness across models of different sizes. It also excels in various downstream tasks, highlighting its strong generalization capabilities.

## Impact Statement

From a scientific perspective, PITA significantly advances physics-informed machine learning by providing a versatile framework that effectively bridges data-driven methodologies with fundamental physical principles. Its capability to discover governing equations directly from data serves as a powerful tool for analyzing complex phenomena, particularly in scenarios where explicit models are incomplete or unavailable. However, it is essential to acknowledge that predictions made by neural networks for physical systems inherently involve approximation errors and often lack interpretability.

PITA shows significant potential for numerous future applications in numerical computing. First, as a development to the traditional auto-regressive approach, PITA can be seamlessly applied to various PDE foundation models, supporting both pretraining and finetuning processes. Second, for domain-specific applications like fluid dynamics, library terms can be tailored according to governing equations such as Navier-Stokes. Thus enables more accurate solutions and better alignment with physical laws. Third, by incorporating PDE discovery, PITA opens up opportunities to utilize other inverse problem-solving techniques, such as initial state recovery and parameter inference, to improve forward problem-solving.

## Acknowledgements

This work was supported in part by the National Natural Science Foundation of China (NSFC) Major Research Plan on Interpretable and General Purpose Next-generation Artificial Intelligence under Grant No. 92370205, NSFC Grant 12425113, the Natural Science Foundation of Jiangsu Province under Grant BK20240462, the China Postdoctoral Science Foundation under Grants 2023M733427 and 2023TQ0349, and the Jiangsu Funding Program for Excellent Postdoctoral Talent. We also acknowledge support from the Key Laboratory of the Ministry of Education for Mathematical Foundations and Applications of Digital Technology, USTC, and the valuable discussions and assistance provided by Zijun Ding.

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

## A. Table of Notations

The primary notations used in this paper are listed in Table 4.

Table 4: Mathematical Notations.

| Notations for Sec. 3.1 | |
|---|---|
| Symbol | Definition |
| $\boldsymbol{x}$ | Spatial variable in the domain $\Omega \subset \mathbb{R}^d$. |
| $\Omega$ | Spatial domain of PDE. |
| $\partial\Omega$ | Boundary of spatial domain $\Omega$. |
| $\boldsymbol{u}(\boldsymbol{x}, t)$ | Time-dependent function defined over the spatial domain $\Omega$ for the time interval $[0, T]$. |
| $X_n$ | Discretized grid of $n$ points in $\Omega$, represented as $\{\boldsymbol{x}_1, \boldsymbol{x}_2, \ldots, \boldsymbol{x}_n\}$, where each $\boldsymbol{x}_i \in \mathbb{R}^d$. |
| $N$ | Total number of spatiotemporal PDEs considered. |
| $d_u$ | Dimensionality of the solution space of $\boldsymbol{u}$. |
| $d_k$ | Dimensionality of the spatial domain for the $k$-th PDE. |
| $\boldsymbol{u}_0(\boldsymbol{x})$ | Initial condition for $\boldsymbol{u}$ at $t = 0$. |
| $\mathcal{F}[\boldsymbol{u}](\boldsymbol{x}, t)$ | Differential operator acting on $\boldsymbol{u}$ and its spatial derivatives. |
| $\mathcal{B}[\boldsymbol{u}](\boldsymbol{x}, t)$ | Boundary condition operator applied at boundary points $\boldsymbol{x} \in \partial\Omega$. |
| $\boldsymbol{u}(\boldsymbol{x}, t)$ | Solution of PDE at time $t$, defined in domain $\Omega$, with values in $\mathbb{R}^{d_u}$. |
| Notations for Sec. 3.2 | |
| Symbol | Definition |
| $\boldsymbol{\theta}$ | Parameters of neural operator. In subsequent sections it refers to the parameters of foundation models. |
| $\mathcal{G}_\theta$ | Neural operator parameterized by weights $\theta$. |
| $T_{in}$ | Roll-in window length, representing the number of input frames used as ground truth. |
| $T_{ar}$ | Roll-out window length, representing the number of frames predicted iteratively by $\mathcal{G}_\theta$. |
| $\boldsymbol{u}_{<T}$ | Set of frames $\{\boldsymbol{u}_i\}_{i=1}^T$ used as input. |
| $\hat{\boldsymbol{u}}_T$ | Predicted frame at time step $T$. |
| $\boldsymbol{u}_T$ | Ground truth frame at time step $T$. |
| Notations for Sec. 3.3 | |
| Symbol | Definition |
| $\|\cdot\|_p$ | $L_p$ norm. |
| $\tilde{X}_w$ | Sampled grid for spatial domain. |
| $w$ | Number of downsampled grid points. |
| $C$ | Number of physical variables. |
| $U$ | Matrix representing the compressed data of $C$ channels across $m$ time points and $n$ spatial locations. |
| $\boldsymbol{\Phi}$ | Library matrix containing $S$ candidate terms and partial derivatives for the PDE. |
| $S$ | Total number of candidate terms in the library. |
| $\boldsymbol{\Lambda}$ | Matrix of the discovered PDE coefficients. |
| $\boldsymbol{\Phi}^\dagger$ | Pseudo-inverse of the library matrix $\boldsymbol{\Phi}$. |

Symbols used in the formulas but not listed in the tables are clarified in the main text.

## B. Illustration of the Shortcut Problem

This section investigates the origin of the shortcut problem and its consequential error accumulation in auto-regressive prediction frameworks. When predicting the $T$-th frame, the prediction error at the ($T$-1)-th time step propagates to subsequent predictions. This propagation triggers cumulative error amplification over successive time steps, which is a critical limitation of auto-regressive mechanisms. As illustrated in Figure 5, within a prediction window of length $T_{ar}$, errors originating from the initial frame amplify iteratively, thereby degrading the accuracy of later predictions. Such accumulation manifests as severe deviations or unphysical artefacts in dynamical system simulations, particularly when predicted states violate the fundamental physical laws represented by the governing PDEs. For instance, in fluid dynamics or climate modeling applications, these artefacts may manifest as non-physical vorticity patterns or temperature distributions that diverge from observed system behavior. The compounding errors not only distort short-term predictions but also erode the validity of extrapolated trajectories over extended timescales. These artefacts ultimately compromise the reliability of

long-term scientific forecasts, raising significant challenges for applications requiring high-fidelity numerical simulations. Furthermore, the error accumulation phenomenon arises from shortcut learning in auto-regressive approaches, where models

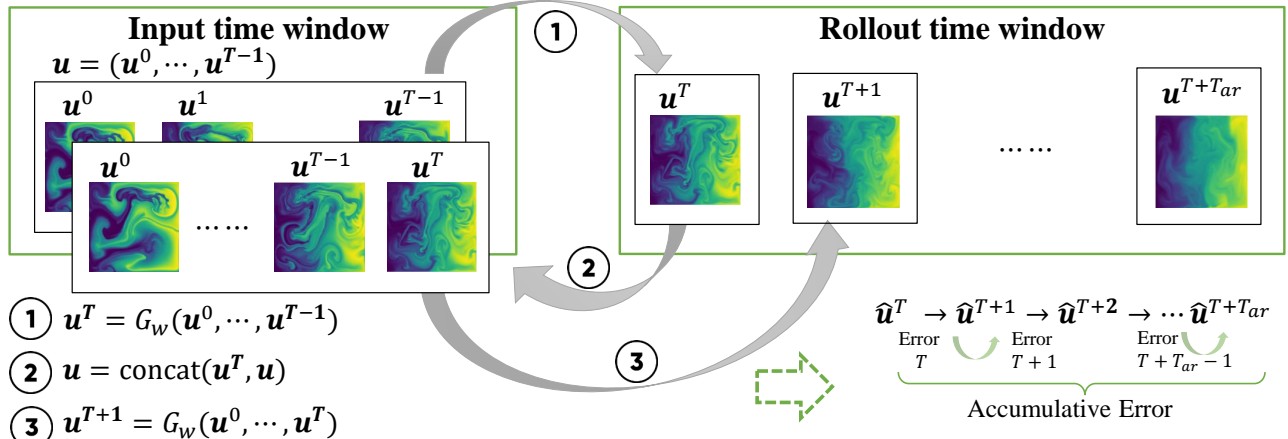

Figure 5: Illustration of Error Propagation in Auto-regressive Prediction.

prioritize copying values from the $(T\text{-}1)$-th frame to predict the $T$-th frame, rather than learning the underlying physical dynamics governed by PDEs. To quantify this behavior, we analyze two long-trajectory datasets using auto-regressive prediction, where the nRMSE is notably higher compared to other methods, as shown in Figure 6. In this figure, the pink curve labeled Temporal SSIM (predicted $t$ vs predicted $t-1$) measures the structural similarity index (SSIM) between consecutive predicted frames, $\hat{u}_{t-1}$ and $\hat{u}_t$. The blue curve labeled Frame SSIM (predicted $t$ vs ground truth $t$ ) quantifies the SSIM between the predicted frame $\hat{u}_t$ and the ground truth frame $u_t$.

At initial time steps, the blue curve dominates the pink curve (Figure 6), indicating that $\hat{u}_t$ aligns more closely with the ground truth $u_t$ than with the prior prediction $\hat{u}_{t-1}$. This suggests the model initially captures physical dynamics encoded in the PDEs rather than relying on superficial shortcut features. However, at later stages, the pink curve surpasses the blue curve, revealing that $\hat{u}_t$ increasingly resembles $\hat{u}_{t-1}$ instead of $u_t$, thereby exacerbating shortcut-driven error propagation. This divergence reflects the model's failure to learn true physical features instead, it recursively replicates prior predictions, introducing incremental errors at each step. Over extended trajectories, the repeated copying of erroneous frame values leads to severe error accumulation, particularly in systems requiring long-term stability, such as turbulence modeling or climate projections. Addressing this limitation is critical for improving the robustness of auto-regressive methods in scientific computing.

## C. Details of Datasets

### C.1. FNO-NS-$\nu$

This benchmark dataset considers the 2D Navier-Stokes equation for a viscous, incompressible fluid in vorticity form on the unit torus. The equation is written as

$$\partial_t w + u \cdot \nabla w = \nu \Delta w + f,$$
$$\nabla \cdot u = 0, \tag{13}$$

where $w$ is vorticity field, $u$ is the velocity field, $f$ is the external force. The only varying component in this dataset is the viscosity coefficient $\nu$, which takes values from the set $\left\{1 \times 10^{-5}, 1 \times 10^{-4}, 1 \times 10^{-3}\right\}$, corresponding to the datasets FNO-NS-1e-5, FNO-NS-1e-4, and FNO-NS-1e-3. For FNO-NS-1e-5, the total length of the testing data is 20 steps, while for both FNO-NS-1e-4 and FNO-NS-1e-3, the length is 30 steps. The task involves predicting future vorticity steps $w(x, t)$ given the initial 10 steps, where $(x, t) \in [0, 1]^2 \times [0, T]$.

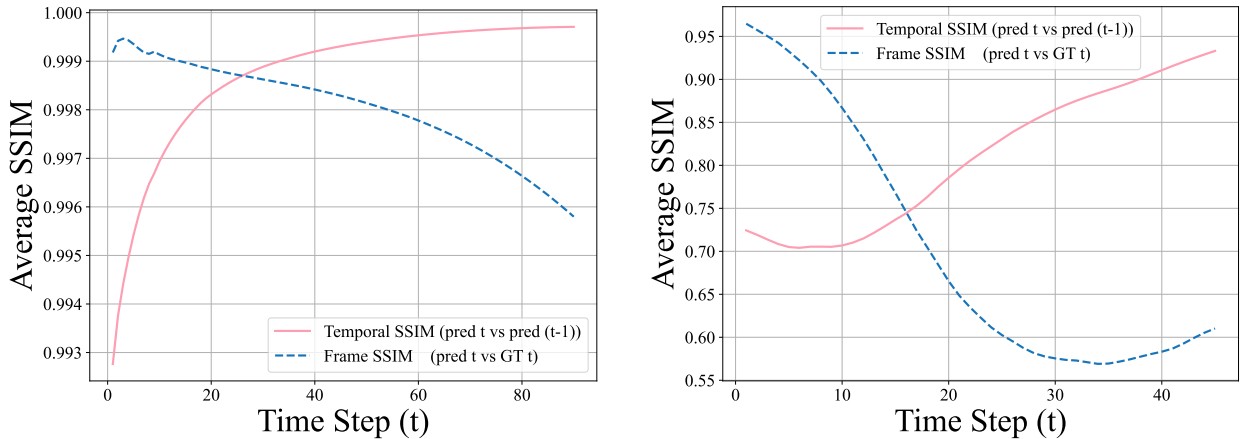

Figure 6: Intrinsic connection between error accumulation and shortcut learning. On the left, we present the shortcut issue on the PDEBench-DR dataset with a trajectory length being 91. On the right, we present the shortcut issue on the NS2D-Force dataset with a trajectory length being 46.

### C.2. PDEBench-CNS-$(M, \eta)$

Different from FNO-$\nu$, this benchmark dataset considers the 2D Navier-Stokes equation for a viscous, compressible fluid. The equation is written as

$$
\begin{aligned}
\partial_t \rho + \nabla \cdot (\rho \mathbf{v}) &= 0, \\
\rho \left( \partial_t \mathbf{v} + \mathbf{v} \cdot \nabla \mathbf{v} \right) &= -\nabla p + \eta \triangle \mathbf{v} + (\zeta + \eta/3) \nabla (\nabla \cdot \mathbf{v}) \\
\partial_t \left[ \epsilon + \frac{\rho v^2}{2} \right] + \nabla \cdot \left[ \left( \epsilon + p + \frac{\rho v^2}{2} \right) \mathbf{v} - \mathbf{v} \cdot \sigma' \right] &= 0,
\end{aligned}
\tag{14}
$$

where $\rho$ represents the mass density, $\mathbf{v}$ is the velocity, $p$ denotes the gas pressure, and $\epsilon = p/(\Gamma - 1)$ is the internal energy, where $\Gamma = 5/3$. The viscous stress tensor is denoted by $\sigma'$, and $\eta, \zeta$ are the shear and bulk viscosity, respectively. The Mach number $M$ is defined by $M = |v|/\sqrt{\Gamma p/\rho}$.

This benchmark examines the varying components of Mach number $M$ and shear viscosity $\eta$ , with values consisting of $(1, 0.1), (1, 0.01), (0.1, 0.1)$, and $(0.1, 0.01)$. The total length of the dataset is 21 steps. The prediction length for the short trajectory is set to 11, whereas for the other datasets, the prediction length for the short trajectory is 10. Given the initial 10 steps, the objective is to predict the velocity $\boldsymbol{v}(\boldsymbol{x}, t)$, pressure $p(\boldsymbol{x}, t)$, and density $\rho(\boldsymbol{x}, t)$ fields, where $(\boldsymbol{x}, t) \in [0, 1]^2 \times [0, 1]$.

### C.3. PDEBench-SWE

The shallow-water equations, derived from the general Navier-Stokes equations, present a suitable framework for modelling free-surface flow problems. The equation is written as

$$
\begin{aligned}
\partial_t h + \partial_x hu + \partial_y hv &= 0, \\
\partial_t hu + \partial_x \left( u^2 h + \frac{1}{2} g_r h^2 \right) + \partial_y uvh &= -g_r h \partial_x b, \\
\partial_t hv + \partial_y \left( v^2 h + \frac{1}{2} g_r h^2 \right) + \partial_x uvh &= -g_r h \partial_y b,
\end{aligned}
\tag{15}
$$

where $u$ and $v$ represent the velocities in the horizontal and vertical directions, $h$ describes the water depth, and $b$ indicates a spatially varying bathymetry. The total length of the testing dataset comprises 101 steps. Given the initial 10 steps, the objective is to predict the water depth $h(\boldsymbol{x}, t)$ within the domain $[-2.5, 2.5]^2 \times [0, 1]$.

## C.4. PDEBench-DR

This benchmark dataset considers diffusion-reaction type PDE that combines a diffusion process and a rapid evolution from a source term. The equation is expressed as

$$\partial_t u = D_u \partial_{xx} u + D_u \partial_{yy} u + R_u, \quad \partial_t v = D_v \partial_{xx} v + D_v \partial_{yy} v + R_v, \tag{16}$$

where $D_u$ and $D_v$ are the diffusion coefficient for the activator and inhibitor, respectively, $R_u = R_u(u, v)$ and $R_v = R_v(u, v)$ are the activator and inhibitor reaction function, respectively. The total length of the testing dataset comprises 101 steps. Given the initial 10 steps, the task is to predict the density fields $u, v$ where the domain is $[-1, 1]^2 \times [0, 5]$.

## C.5. PDEArena

This benchmark dataset focuses on the incompressible Navier-Stokes equations in velocity function and vorticity stream formulation,

$$\frac{\partial \boldsymbol{v}}{\partial t} = -\boldsymbol{v} \cdot \nabla \boldsymbol{v} + \mu \nabla^2 \boldsymbol{v} - \nabla p + \boldsymbol{f},$$
$$\nabla \cdot \boldsymbol{v} = 0, \tag{17}$$

where $\boldsymbol{v}$ represents the velocity flow fields, $\boldsymbol{f}$ denotes the external force, and $p$ is the internal pressure. This benchmark consists of two subsets: the PDEArena-NS, which includes a fixed external force, and the PDEArena-NS-Force, which incorporates a variable external force. For the PDEArena-NS dataset, the length of the testing data is 14 steps, while for the PDEArena-NS-Force dataset, the length is 56 steps. The objective for both datasets is to predict the velocity $\boldsymbol{v}(\boldsymbol{x}, t)$, pressure $p(\boldsymbol{x}, t)$, and density $\rho(\boldsymbol{x}, t)$ fields using the initial 10 steps, where $(\boldsymbol{x}, t) \in [0, 32]^2 \times [0, 24]$.

## C.6. CFDBench-NS

This benchmark dataset focuses on the incompressible fluid dynamics on domains with irregular geometries

$$\frac{\partial}{\partial t}(\rho \boldsymbol{u}) + \nabla \cdot (\rho \boldsymbol{u}^2) = -\nabla p + \nabla \cdot \mu \left[ \nabla \boldsymbol{u} + (\nabla \boldsymbol{u})^\top \right],$$
$$\nabla \cdot (\rho \boldsymbol{u}) = 0, \tag{18}$$

where $\rho$ is the density and $\mu$ is the dynamic viscosity, $\boldsymbol{u}$ is the velocity field, and $p$ is the pressure. The total length of the testing dataset consists of 20 steps. The task is to predict the velocity field $\boldsymbol{u}(\boldsymbol{x}, t)$ given the initial 10 time steps.

## C.7. Magnet-Viscous Burgers

To evaluate whether PITA can effectively handle downstream tasks, we consider the Burgers' equation, which models the dynamics of a field $\boldsymbol{u}$ incorporating nonlinear advection and diffusion. The equation is written as:

$$\partial_t \boldsymbol{u} + \boldsymbol{u} \nabla \boldsymbol{u} = \beta \Delta \boldsymbol{u},$$
$$\boldsymbol{u}(0, x, y) = \sum_{j=1}^{5} A_j \sin\left(2\pi l_j^x x/64 + \varPhi_j^x\right) \cos\left(2\pi l_j^y y/64 + \varPhi_j^y\right), \tag{19}$$

where $\boldsymbol{u}$ represents the field, $\beta$ is the diffusion coefficient $\beta \in (0, 0.2]$, and the coefficients of initial conditions are sampled as $A_j \in [-0.5, 0.5]$, $l_j^x, l_j^y \in \{1, 2, 3\}$, and $\varPhi_j^x, \varPhi_j^y \in [0, 2\pi)$. The total length of the testing dataset consists of 50 steps. The task is to predict the field $\boldsymbol{u}(\boldsymbol{x}, t)$ over the domain $(\boldsymbol{x}, t) \in [0.25, 63.75]^2 \times [0, 9.8]$, given the initial 10 steps.

# D. Training Details

## D.1. Model Configuration

Below, we provide the details of the various model configurations and scales used in the paper.

- **DPOT:** In Table 5, we provide the training configurations of DPOT (Hao et al., 2024).

Table 5: Configurations of DPOT of different sizes.

| Size | Attention Dim | MLP Dim | Layers | Heads | Params |
|------|---------------|---------|--------|-------|--------|
| Tiny | 512 | 512 | 4 | 4 | 7M |
| Small | 1024 | 1024 | 6 | 8 | 30M |
| Medium | 1024 | 4096 | 12 | 8 | 122M |
| Large | 1536 | 6144 | 24 | 16 | 509M |

- **FNO:** In Table 6, we provide the training configurations of FNO (Li et al., 2020), where Mode1 and Mode2 represent the number of Fourier modes used in $x$ and $y$ spatial dimension respectively.

Table 6: Configurations of FNO.

| Size | Mode1 | Mode2 | Width | Depth | Params |
|------|-------|-------|-------|-------|--------|
| Medium | 8 | 5 | 512 | 4 | 170M |

- **MPP:** In Table 7, we provide the training configurations of MPP (McCabe et al., 2023).

Table 7: Configurations of MPP.

| Size | Embed Dim | MLP Dim | Heads | Blocks | Patch Size | Params |
|------|-----------|---------|-------|--------|------------|--------|
| Tiny | 192 | 768 | 3 | 12 | [16,16] | 7.6M |
| Small | 384 | 1536 | 6 | 12 | [16,16] | 29M |

### D.2. Hyperparameters

The following training hyperparameters are used across all experiments, except where explicitly stated otherwise.

Table 8: Training Hyperparameter Settings Across Models and Strategies.

| Model | DPOT | | MPP | | FNO | |
|-------|------|------|-----|------|-----|------|
| Strategy | Auto-regressive | PITA | Auto-regressive | PITA | Auto-regressive | PITA |
| Batch Size | 20 | 20 | 24 | 24 | 20 | 20 |
| Gradient Clipping | 10000 | 10000 | 1 | 1 | 1 | 1 |
| Dropout | 0 | 0 | 0.1 | 0.1 | 0 | 0 |
| Initial Learning Rate | 1e-3 | 1e-3 | 1e-3 | 1e-3 | 1e-3 | 1e-3 |
| Optimizer | Adam | Adam | AdamW | AdamW | Adam | Adam |
| Learning Rate Schedule | Cycle | Cycle | Cycle | Cycle | Cycle | Cycle |
| Weight Decay | 1e-6 | 1e-6 | 5e-2 | 5e-2 | 1e-6 | 1e-6 |
| Warmup Epoch | 50 | 50 | 5 | 5 | 50 | 50 |
| optimizer momentum | (0.9,0.9) | (0.9,0.9) | (0.9,0.999) | (0.9, 0.999) | (0.9,0.9) | (0.9,0.9) |

## E. Supplementary Experimental Results

### E.1. Data Efficiency Analysis of Downstream Tasks

To establish that PITA can be effectively applied to downstream tasks with limited and costly data acquisition, we conducted a series of experiments with varying sample sizes, analyzing data efficiency as shown in Table 9. Our experiments utilize the DPOT-S model. In general, both PITA and the auto-regressive model exhibit improved performance with an increased number of samples, whether training from scratch or finetuning from pretrained models. Notably, when finetuning from pretrained models, PITA achieves comparable prediction accuracy to the auto-regressive model, utilizing only 500

samples—half the number required by the latter, which requires 1000 samples to reach a similar level of accuracy. Moreover, when training from scratch, PITA demonstrates state-of-the-art performance, achieving an average reduction in nRMSE by 11.48%. This significant improvement highlights PITA's efficiency and suitability for scenarios where data is scarce and expensive to obtain.

Table 9: Comparisons of Data Efficiency on Downstream Tasks.

| Samples Size | Finetuning from Pretrained Model | | Training from Scratch | |
|---|---|---|---|---|
| | Auto-regress | PITA | Auto-regress | PITA |
| 10 | 0.1092 | 0.10273 | 0.14395 | 0.13873 |
| 100 | 0.01077 | 0.00998 | 0.01538 | 0.01405 |
| 500 | 0.00549 | 0.00514 | 0.00640 | 0.00586 |
| 800 | 0.00543 | 0.00451 | 0.00636 | 0.00531 |
| 1000 | 0.00518 | 0.00504 | 0.00629 | 0.00502 |

### E.2. Performance Comparison of PITA and Auto-regressive Methods on Long Trajectory Datasets

To demonstrate that PITA effectively helps the shortcut problem commonly encountered in long trajectory datasets and achieves superior performance, we present the prediction results for both auto-regressive and PITA methods in Figure 7. The nRMSE for each method across the datasets is calculated as the average of the prediction results from all models, as detailed in Table 1. The results indicate that PITA consistently outperforms other methods across all long trajectory datasets, achieving an average nRMSE reduction of 29.15%. Except for the PDEBench-DR and PDEArena-NS-Force datasets, PITA also significantly outperforms the auto-regressive training approach, achieving an average nRMSE reduction of 15.93%. While the performance of PITA on the two aforementioned datasets is relatively close to that of auto-regressive, the results remain competitive.

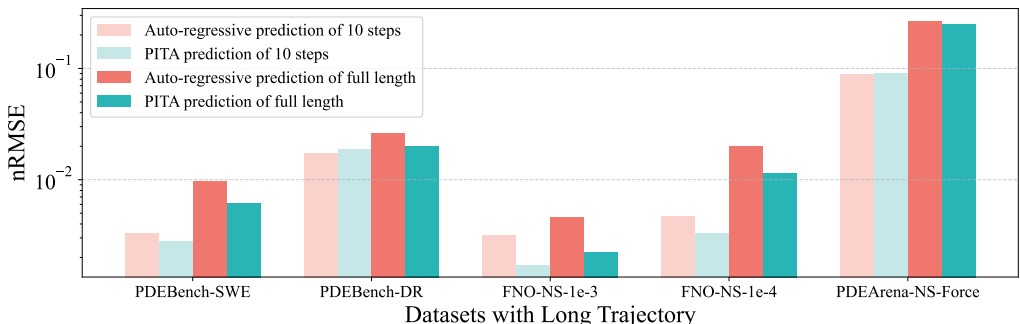

Figure 7: Results of PITA and Auto-Regressive Methods on Long Trajectory Datasets.

### E.3. Scaling Experiments

In this section, we analyze the scaling results from Table 1, which are illustrated in Figure 8. It is observed that as the model size increases, the average nRMSE across the five datasets with long trajectories decreases, generally following a scaling law. However, on two larger-scale datasets (FNO-NS-1e-3 and FNO-NS-1e-4) PITA performs less stably on the larger-sized baselines. The main reason may be that the DPOT baseline uses a cyclic learning rate decay incorporating very high learning rates during training. This may lead to gradient explosion or overly aggressive weight updates, affecting the convergence of models with more parameters. Furthermore, DPOT performs finetuning for all model sizes with only 500 epochs and sets the gradient clipping value to 10,000. This prevents PITA from converging stably on larger datasets with larger model sizes. We found that this issue can be mitigated after adjusting the training settings. For example, when we provide sufficient training, PITA shows more stable convergence results, as shown in Table 10 of Appendix E.4. To demonstrate PITA's plug-and-play performance more fairly, we still follow DPOT's training settings in the experiments, without making any adjustments to PITA, except for special declarations.

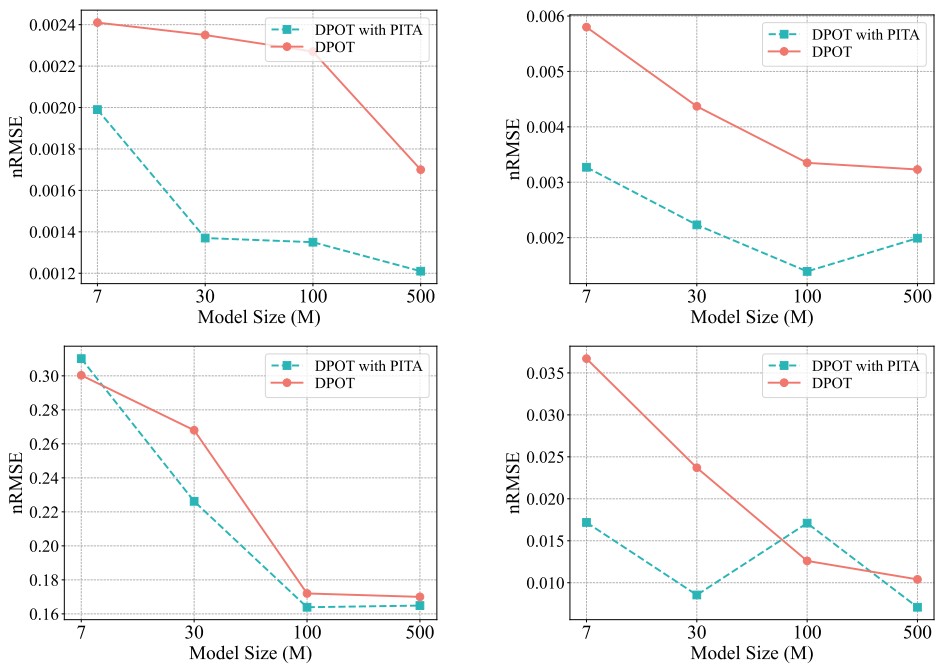

Figure 8: Results of scaling experiments for different dataset sizes. In the first row, we present results for PDEBench-SWE (left) and FNO-NS-1e-3 (right). In the second row, we show results for NS-Force (left) and FNO-NS-1e-4 (right).

### E.4. Comparative Analysis of Training from Scratch and Finetuning Pretrained Models

To compare the performance of pretrained models with those trained from scratch, we selected four datasets, as shown in Table 10. Testing on the PDEBench-DR and FNO-NS-1e-5 datasets represents in-distribution testing, while the PDEArena-SWE and Burgers serve as the out-of-distribution test. It is noteworthy that although the shallow-water equation was included in the training dataset for the DPOT model, the only physics variable observed during training was the water depth. In the out-of-distribution context, in addition to predicting the surface height $h$, the model will also predict 88 steps of zonal velocity $u$, meridional velocity $v$, pressure field $p$, and wind vorticity field $\xi$.

Firstly, for in-distribution test, the results from finetuning are slightly better than those achieved by training from scratch using the auto-regressive and PITA methods. This improvement is attributed to the former's training over 2500 epochs, compared to only 1500 epochs for the latter. With a reduction of 19.684% in nRMSE, the training cost for finetuning is 1.5 times that of training from scratch using PITA or the auto-regressive method. This suggests that large scale pretraining on PDE datasets may not be effective and does not scale well with an increased number of training epochs. Secondly, in out-of-distribution test, the results from finetuning using pretrained models are generally worse than those from training from scratch. This indicates that during the pretraining process, the model primarily learns the physical laws from the training datasets, lacking the capacity to generalize to unseen tasks where the underlying physical principles are unknown. Moreover, the physics laws governing different families of PDEs share little similarity, even among PDEs within the same family but with varying parameters. As a result, a foundation model may struggle on out-of-distribution tests. Lastly, when training from scratch, the results are generally better than those obtained using the auto-regressive method. This observation highlights PITA's potential application in PDE foundation models.

### E.5. Comparison of PITA and Auto-regressive Methods ($T_{ar} = 10$)

In this section, we present the training results for $T_{ar} = 10$. Each dataset's full trajectory is predicted, with the roll-out prediction length fixed at $T_{ar} = 10$. Consequently, Table 11 contains only the results for long trajectory predictions. It is observed that when the roll-out window length is set to 10, the results are generally poorer compared to those from $T_{ar} = 1$. This decline in performance can be attributed to the shortcut problem that occurs during training.

Table 10: Experimental Results of Pretrained Models Versus Models Trained from Scratch Using Auto-regressive and PITA Methods. Finetune in the table indicates that the model is finetuned on a pretrained model of a specific size. Train from Scratch (Auto-regress) refers to models trained using the auto-regressive approach without any pretrained models. Train from Scratch (PITA) denotes models trained using the PITA approach without any pretrained models.

| Model | Epoch / Strategy | In Distribution | | | | | | Out of Distribution | | | | | |
| | | PDEBench-DR | | | FNO-NS-1e-5 | | | PDEArena-SWE | | | Burgers | | |
| | | 500 | 1000 | 1500 | 500 | 1000 | 1500 | 500 | 1000 | 1500 | 500 | 1000 | 1500 |
|---|---|---|---|---|---|---|---|---|---|---|---|---|---|
| DPOT-Ti | Finetune (Auto-regress) | 0.12281 | 0.04249 | 0.00522 | 0.11144 | 0.06398 | 0.03595 | 18.46497 | 5.24994 | 1.01142 | 0.03712 | 0.02543 | 0.00455 |
| | Train from Scratch (Auto-regress) | 0.29705 | 0.05467 | 0.00651 | 0.18219 | 0.07222 | 0.04072 | 5.98731 | 3.20792 | 0.99990 | 0.06691 | 0.01526 | 0.00419 |
| | Train from Scratch (PITA) | 0.42291 | 0.03921 | 0.00577 | 0.15501 | 0.07312 | 0.04009 | 2.07097 | 3.89625 | 0.99710 | 0.06945 | 0.01318 | 0.00374 |
| DPOT-S | Finetune (Auto-regress) | 0.14761 | 0.02113 | 0.00456 | 0.11242 | 0.04399 | 0.03054 | 44.70683 | 2.69744 | 0.99152 | 0.04679 | 0.01627 | 0.00304 |
| | Train from Scratch (Auto-regress) | 0.20054 | 0.05483 | 0.00617 | 0.11495 | 0.04570 | 0.03771 | 8.0087 | 2.66849 | 0.99414 | 0.06251 | 0.02703 | 0.00383 |
| | Train from Scratch (PITA) | 0.80244 | 0.05853 | 0.00807 | 0.11741 | 0.06509 | 0.03701 | 16.16088 | 1.87741 | 0.99051 | 0.04121 | 0.01943 | 0.00307 |

Table 11: Comparisons of PITA and Auto-regressive Strategies Across Various Models and Datasets ($T_{ar} = 10$).

| Model | Strategy | FNO-NS-$\nu$ | | | PDEBench | | PDEBench CNS-$(\eta, \zeta)$ | | | | | | PDEArena | | CFDBench |
| | | 1e-5 | 1e-4 | 1e-3 | DR | SWE | 1,0.1 | 1,0.01 | M1 | 0.1,0.1 | 0.1,0.01 | M0.1 | NS-Force | NS | |
|---|---|---|---|---|---|---|---|---|---|---|---|---|---|---|---|
| | | | | | | | *Small Models* | | | | | | | | |
| DPOT-Ti 7M | Auto-regress | 0.0982 | 0.0779 | 0.0065 | 0.0342 | 0.0083 | 0.0523 | 0.0345 | 0.0434 | **0.0492** | 0.0213 | **0.0352** | 0.3441 | 0.0762 | 0.0027 |
| | PITA | **0.0897** | **0.0692** | **0.0058** | **0.0286** | **0.0032** | **0.0505** | **0.0298** | **0.0402** | 0.0531 | **0.0262** | 0.0397 | **0.2862** | **0.0585** | **0.0019** |
| DPOT-S 30M | Auto-regress | 0.0879 | 0.0406 | 0.0022 | 0.0312 | 0.0036 | 0.0678 | 0.0280 | 0.0479 | 0.0439 | 0.0357 | 0.0398 | 0.2185 | 0.0570 | 0.0028 |
| | PITA | **0.0790** | **0.0371** | 0.0025 | **0.0183** | **0.0016** | 0.0548 | 0.0301 | 0.0425 | **0.0405** | **0.0311** | 0.0358 | 0.2335 | 0.0621 | **0.0016** |
| DPOT-M 122M | Auto-regress | 0.0785 | 0.0402 | 0.0031 | 0.0384 | 0.0032 | 0.0512 | 0.0216 | 0.0364 | **0.0517** | 0.0216 | 0.0367 | 0.2256 | 0.0540 | 0.0039 |
| | PITA | **0.0630** | **0.0285** | **0.0019** | **0.0305** | **0.0020** | 0.0468 | 0.0175 | 0.0322 | 0.0532 | 0.0208 | 0.0370 | **0.1878** | **0.0500** | **0.0028** |
| FNO-M 170M | Auto-regress | 0.1592 | 0.0338 | 0.0029 | - | 0.0023 | 0.0527 | 0.0373 | 0.0450 | 0.0588 | 0.0263 | 0.0425 | 0.3064 | 0.2446 | 0.0041 |
| | PITA | **0.1300** | **0.0148** | **0.0018** | - | 0.0030 | **0.0478** | 0.0414 | **0.0446** | 0.0621 | 0.0310 | 0.0466 | **0.2485** | **0.1997** | 0.0049 |
| | | | | | | | *Large Models* | | | | | | | | |
| DPOT-L 500M | Auto-regress | 0.0704 | 0.0208 | **0.0031** | 0.0344 | **0.0018** | 0.0308 | 0.0344 | 0.0326 | 0.0585 | **0.0287** | 0.0436 | 0.1741 | 0.0562 | **0.0036** |
| | PITA | **0.0688** | **0.0185** | 0.0036 | **0.0299** | 0.0019 | **0.0274** | 0.0328 | **0.0301** | 0.0523 | 0.0312 | **0.0417** | **0.1675** | **0.0509** | 0.0039 |

### E.6. Comparison of Sparse Regression and Other Numerical Algorithms

The choose of sparse regression framework for PDE discovery is primarily based on its theoretical and experimental advantages. Firstly, the $\ell_0$ regularization in sparse regression in our PDE discovery step inherently suppresses noise by enforcing sparsity in the candidate coefficient space. As noted in compressed sensing theory (Donoho, 2006), sparse regularization, particularly $\ell_0$ regularization, prunes small-magnitude terms caused by observational noise, effectively driving them to zero while preserving dominant dynamical terms (Huang & Aviyente, 2006; Wen et al., 2018). Additionally, prior work (Chen et al., 2021; Rudy et al., 2017) has provided experimental evidence that sparse regression can successfully identify governing equations with high accuracy even in the presence of noisy data. The $\ell_0$-norm regularization adopted in this paper systematically eliminates noise-dominated low-magnitude terms through a hard thresholding mechanism.

Secondly, compared to the sparse regression algorithm, existing regression methods exhibit significant limitations in complex correlation and noise scenarios. While the classical LASSO (Tibshirani, 1997) method achieves sparsity through $\ell_1$ regularization, its convex relaxation property tends to randomly select collinear features when high correlations exist between columns of the data matrix (as demonstrated in the typical PDE identification scenario in reference (Knowles & Renka, 2014)), leading to unstable identification of true physical patterns and resulting in non-unique solutions and pseudo-sparsity phenomena. The sequentially thresholded least squares (STLS) method (Budišić et al., 2012) improves sparse identification through a recursive thresholding mechanism, but its iterative process based on ordinary least squares lacks regularization mechanisms. In the presence of ill-conditioned matrices or highly correlated features (as shown in the experiments of reference (Budišić et al., 2012)), the computation of unregularized inverse matrices amplifies noise sensitivity and leads to cumulative parameter estimation bias with increasing iterations. Regarding interpretability, although traditional ridge regression improves matrix condition numbers and mitigates collinearity effects through $\ell_2$ regularization (as theoretically analyzed in reference (Noack et al., 2003)), its inherent non-sparse solution property fundamentally conflicts with the interpretability requirements of physical models.

Thirdly, in our experiments, when employing other non-sparse methods such as pseudo-inverse or least squares regression (Golub & Pereyra, 1973), the resulting coefficient matrices exhibit significant non-physical oscillations due to the lack of sparsity constraints in the solution space. This leads to an increase in the second-order norm of the residual terms by 2 or 3 orders of magnitude. Such ill-conditioned solutions make it difficult to effectively normalize $\mathcal{L}_{Phy}$ and can also trigger gradient explosion phenomena during back propagation.

Table 12: Comparisons of Numerical Algorithms Adopted for PITA in the DPOT-Ti Model.

|                      | PDEBench-DR         | FNO-NS-1e-5         |
|----------------------|---------------------|---------------------|
| Numerical Algorithm  | Test Loss (91 Steps)| Test Loss (10 Steps)|
| Pseudoinverse        | 0.02016             | 0.08971             |
| Least Squares Method | 0.06903             | 0.09470             |
| Sparse Regression    | 0.01090             | 0.05629             |

## E.7. Comparison of Computational Cost Between PITA and Auto-regressive Methods

PITA differs from conventional auto-regressive methods by incorporating a PDE discovery procedure, which introduces additional training-time computation and CPU memory usage. We measured per-batch training time, the extra time incurred by the PDE discovery process ("STRidge time" in Table 13), and CPU memory consumption. Given that GPU memory usage remains invariant relative to the baseline, it is therefore excluded from our measurements. The results are summarized in Table 13 and Figure 9.

As shown in Figure 9 (right), the overhead due to PDE discovery (detailed in Section 3.3) is constant and independent of model size. During inference, PITA matches the baseline's efficiency—because discovery and alignment occur only during training—and consistently outperforms it on long-term prediction benchmarks (see Figure 7).

Figure 9 (left) shows that PITA requires no additional GPU memory but incurs a fixed, moderate increase in CPU memory for candidate library construction and sparse regression. This decoupling of computational overhead from model dimensionality enables PITA to scale to billion-parameter architectures without prohibitive resource demands. For example, a 500 M-parameter, L-level model incurs only 0.517s extra per batch ( 20% increase) while achieving a 31.61% improvement in long-trajectory prediction accuracy.

Table 13: Computation Cost Comparison Between PITA and Auto-regressive.

| Size | Model           | Training Time (s) | STRidge Time (s) | CPU Memory Consumption (MB) |
|------|-----------------|-------------------|------------------|-----------------------------|
| Ti   | PITA            | 0.4124            | 0.0959           | 1429.43                     |
|      | Auto-regressive | 0.2941            | -                | 1060.53                     |
| S    | PITA            | 0.5783            | 0.0847           | 1533.21                     |
|      | Auto-regressive | 0.3464            | -                | 1095                        |
| M    | PITA            | 1.2012            | 0.1124           | 2207.5                      |
|      | Auto-regressive | 0.8924            | -                | 1937.48                     |
| L    | PITA            | 3.0381            | 0.1098           | 2581.84                     |
|      | Auto-regressive | 2.5207            | -                | 2203.63                     |

## E.8. Influence of the Percentage of Remained Library Terms

To evaluate the impact of the incomplete library, we conducted experiments with subsampled candidate term sets, as shown in Table 14. When randomly retaining 50% of the library terms, the test loss on PDEBench-SWE increases from a baseline of 0.00137 (achieved with full-term libraries) to 0.00213. Notably, this performance degradation diminishes when preserving 80% of library terms, where the test loss stabilizes at 0.00135 – statistically comparable to the full-library configuration. Crucially, this finding indicates that the library architecture requires no specialized customization for individual PDE systems.

Table 14: Influence of the Percentage of Remained Library Terms on Test Loss.

| Percentage of the Remained Library Terms | Test Loss (10 Steps) | Test Loss (Full Length) |
|------------------------------------------|----------------------|-------------------------|
| 50%                                      | 0.00209              | 0.00213                 |
| 80%                                      | 0.00145              | 0.00135                 |
| 100%                                     | 0.00150              | 0.00137                 |

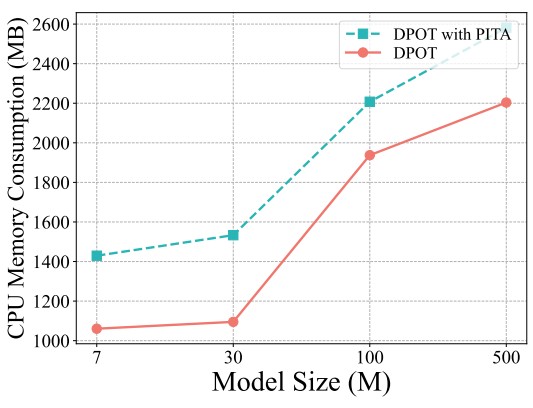 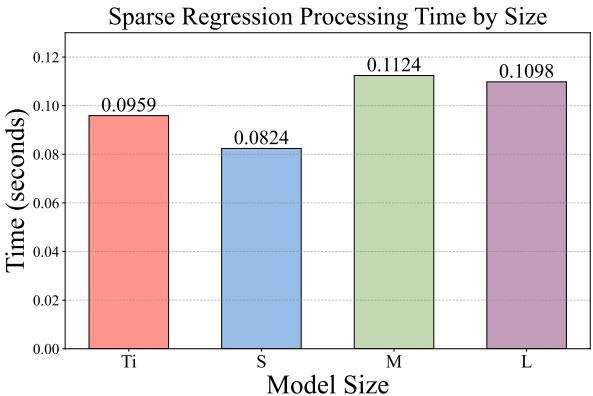

Figure 9: Results of Computational Costs Between PITA and Auto-regressive Method. In the left, we present the CPU memory consumption of each approach. In the right, we show the additional computational time for the sparse regression process of PITA.

### E.9. Influence of Accuracy of the Discovered Physics Accuracy on PITA's Performance

In this discussion, we designed a specific experiment to evaluate PITA's robustness against dynamic failure, where we deliberately injected noise into the gradient computation of PITA's physics loss to simulate inaccurate or corrupted dynamics. As illustrated in Table 15, even under such a direct attack on the physics constraint, PITA still exhibits strong robustness.

Table 15: Influence of Discovered Physics Accuracy on PITA Under Noise Attack in Terms of Physics Loss.

| | PDEBench-SWE | |
|---|---|---|
| Method | nRMSE (10 Steps) | nRMSE (91 Steps) |
| PITA (Salt Pepper Noise) | 0.00224 | 0.00256 |
| PITA (Without Noise) | 0.00202 | 0.00199 |
| Auto-regressive | 0.00208 | 0.00241 |

We further validate robustness by perturbing the PDE spatial grid (see Table 16), showing that even with an intentionally distorted grid, the method maintains performance parity with unperturbed settings. Even when the PDE discovery fails, the multi-task learning framework mitigates its negative impact, ensuring that the model degrades gracefully to the baseline performance.

Table 16: Influence of Discovered Physics Accuracy on PITA in the DPOT-Ti Model.

| | PDEBench-SWE | | PDEBench-DR | |
|---|---|---|---|---|
| Grid Type | nRMSE (10 Steps) | nRMSE (91 Steps) | nRMSE (10 Steps) | nRMSE (91 Steps) |
| Wrong Grid | 0.0024 | 0.00231 | 0.02115 | 0.01684 |
| Correct Grid | 0.00202 | 0.00199 | 0.02119 | 0.0109 |

### E.10. Performance on Modeling Chaotic Dynamical Systems

To rigorously evaluate PITA's capacity for modeling chaotic dynamical systems under extended temporal extrapolation, we conducted controlled experiments on synthetic 2D Kolmogorov turbulence flows, which is a canonical benchmark for chaotic PDE systems exhibiting multiscale energy cascades and nonlinear dissipation processes. Our turbulence simulation dataset contains 300 spatiotemporal trajectories with $64 \times 64$ grid resolution and 150 temporal steps, intentionally designed to challenge long-term forecasting fidelity. Both methodologies leverage the same pretrained DPOT-Ti model. As shown in Table 17, our results demonstrate that PITA significantly outperforms the auto-regressive baseline in capturing the intricate patterns of chaotic fluid evolution. Specifically, PITA exhibits superior alignment with the underlying physical dynamics, showcasing its enhanced ability to model complex, long-term behaviors in turbulent systems.

Table 17: Test Loss on Kolmogorov Turbulence Flow.

|  | Test Loss (10 Steps) | Test Loss (150 Steps) |
|---|---|---|
| Auto-regressive | 0.2581 | 0.4459 |
| PITA | 0.2247 | 0.3152 |

### E.11. Robustness Analysis of PITA under Noise and Data Masking

One of the most significant challenges facing PDE foundation models is their ability to cope with noisy or incomplete data in real-world scenarios. To assess PITA's robustness, we firstly conduct a series of experiments in which Gaussian noise is injected into the dataset at three different levels: 0.05, 0.005 and 0.0005. As summarized in Table 18, PITA consistently outperforms all baseline methods under every noise condition, achieving an average accuracy improvement of 51.79 %. Notably, even at the highest noise level of 0.05, PITA maintains stable performance, demonstrating a much smaller degradation compared to its peers. These results not only confirm PITA's strong resilience to data corruption but also highlight its practical potential for deployment in applications where high-quality measurements are difficult to obtain.

Table 18: Test Loss with Noisy Data at Different Noise Levels on the DPOT-Ti Model with the FNO-NS-1e-3 Dataset.

|  | Auto-regressive | | PITA | |
|---|---|---|---|---|
| Noise Scale | nRMSE (10 Steps) | nRMSE (20 Steps) | nRMSE (10 Steps) | nRMSE (20 Steps) |
| 0.05 | 0.32331 | 0.45795 | 0.2017 | 0.3214 |
| 0.005 | 0.01947 | 0.0631 | 0.00828 | 0.01241 |
| 0.0005 | 0.00402 | 0.00608 | 0.00237 | 0.00333 |

Secondly, we evaluate PITA's capability to reconstruct solutions from incomplete observations. Specifically, 25% of the PDE dataset is randomly masked, and the model is tasked with predicting the full field, including within the occluded regions. As reported in Table 19, PITA significantly outperforms the auto-regressive baseline under these conditions. While all methods experience some performance degradation when confronted with missing data, PITA's relative drop in accuracy is markedly lower. We attribute this resilience to PITA's integrated PDE-discovery module, which infers the governing dynamics even from downsampled, partially missing inputs, enabling it to generalize beyond the observed measurements. These results demonstrate PITA's enhanced ability to recover accurate solutions in data-sparse settings, highlighting its potential for deployment in real-world scenarios where measurement gaps are ubiquitous.

Table 19: Test Loss with Incomplete Data on the DPOT-Ti Model with the PDEBench-SWE Dataset.

|  | Auto-regressive | | PITA | |
|---|---|---|---|---|
| Mask Ratio | nRMSE (10 Steps) | nRMSE (91 Steps) | nRMSE (10 Steps) | nRMSE (91 Steps) |
| 0.25 | 0.02176 | 0.1264 | 0.0112 | 0.0801 |

### E.12. Error Bars of PITA on Long Trajectory Datasets

To ascertain that the improvements of PITA demonstrated in Table 1 stem specifically from its PDE discovery framework and alternating direction optimization algorithm, we evaluate the performance of PITA against auto-regressive methods on datasets featuring extended temporal trajectories. Such datasets inherently pose challenges for long-term prediction and downstream tasks due to their temporal complexity. By assessing multiple model scales, we further investigate how variance patterns and reliability metrics correlate with model complexity. As illustrated in Figure 10, the results confirm that PITA's enhancements are statistically significant and intrinsically tied to its architectural design, rather than stochastic fluctuations.

## F. Visualizations of Trajectory

In this section, we visualize datasets predictions presented in Table 1.

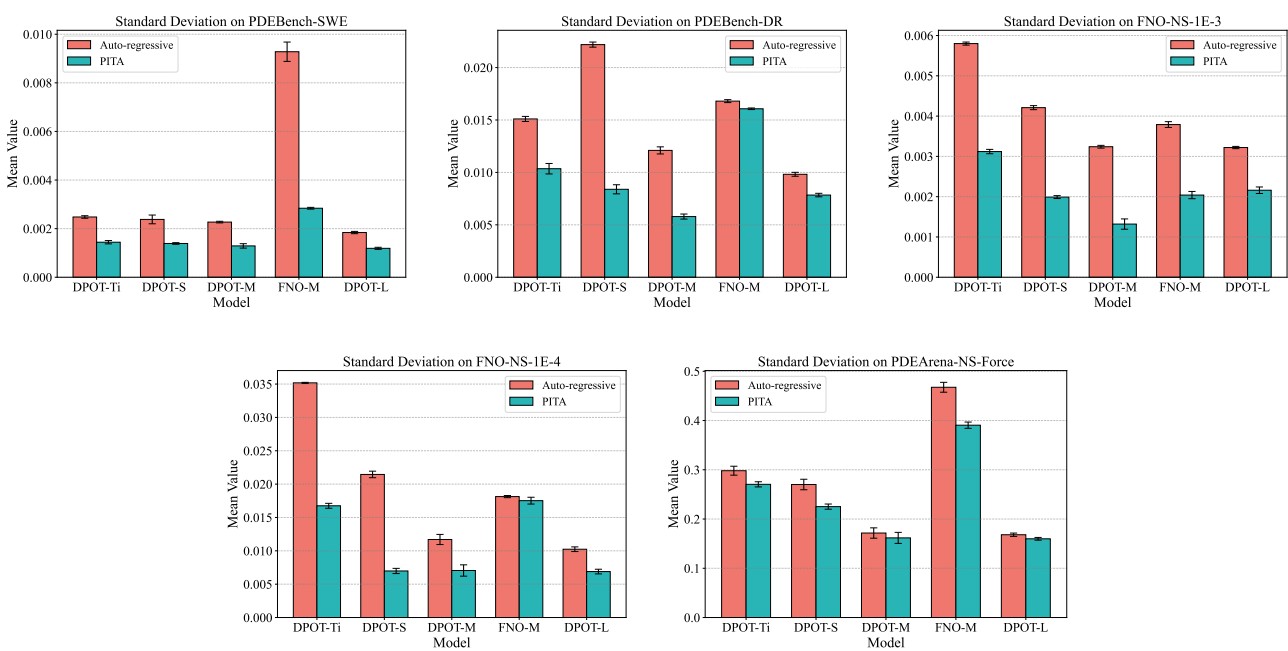

Figure 10: Mean Value and Standard Deviation on Datasets with Long Trajectory.

## F.1. PDEBench-DR

In this section, we visualize two quantities from the diffusion-reaction equation: the density fields $u$ and $v$.

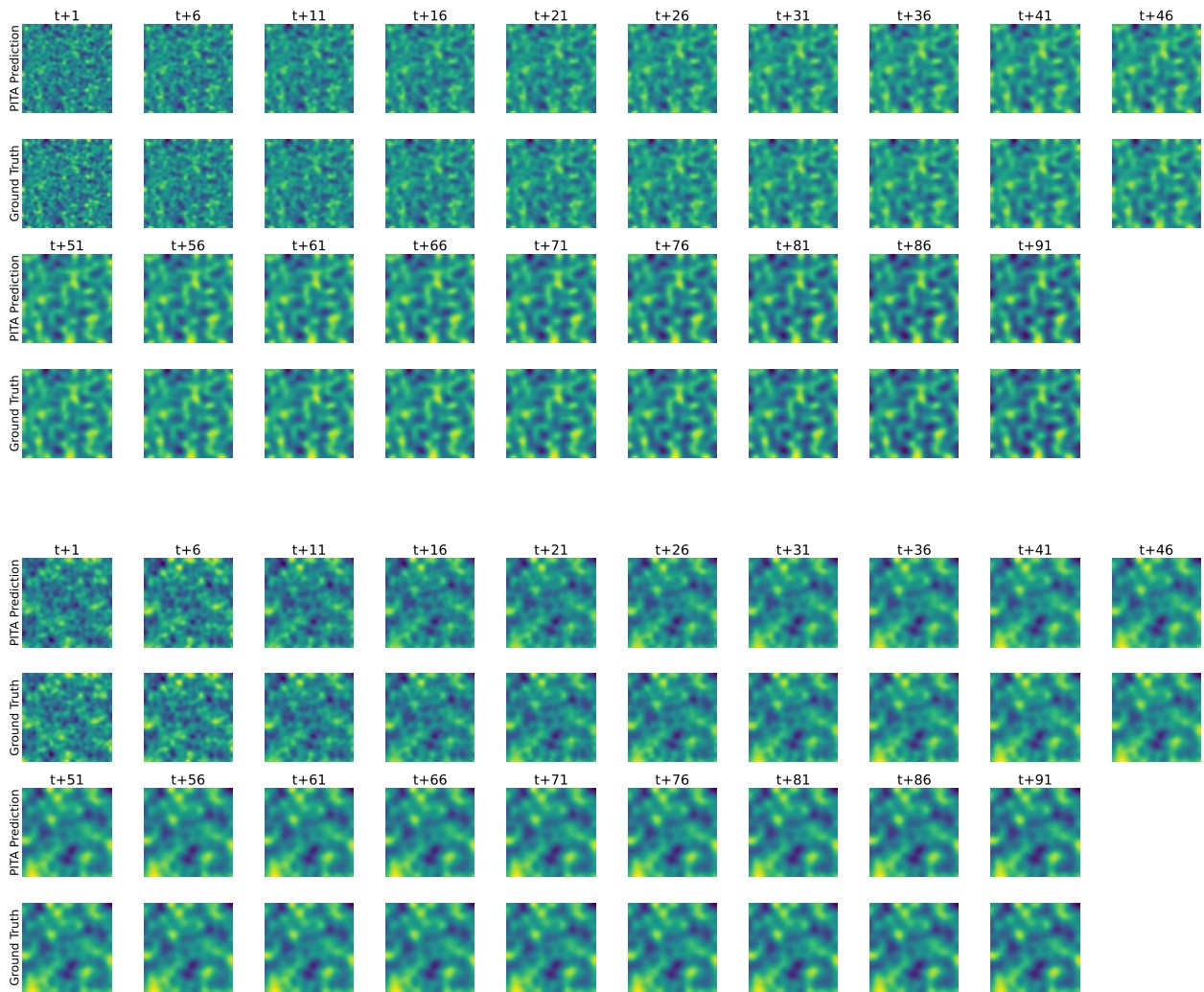

## F.2. FNO-NS-$\nu$

Here we visualize the vorticity $w$ of the incompressible Navier-Stokes equation, with different viscous coefficient $\nu = 1 \times 10^{-3}, 1 \times 10^{-4}, 1 \times 10^{-5}$.

- $\nu = 1 \times 10^{-3}$

- $\nu = 1 \times 10^{-4}$

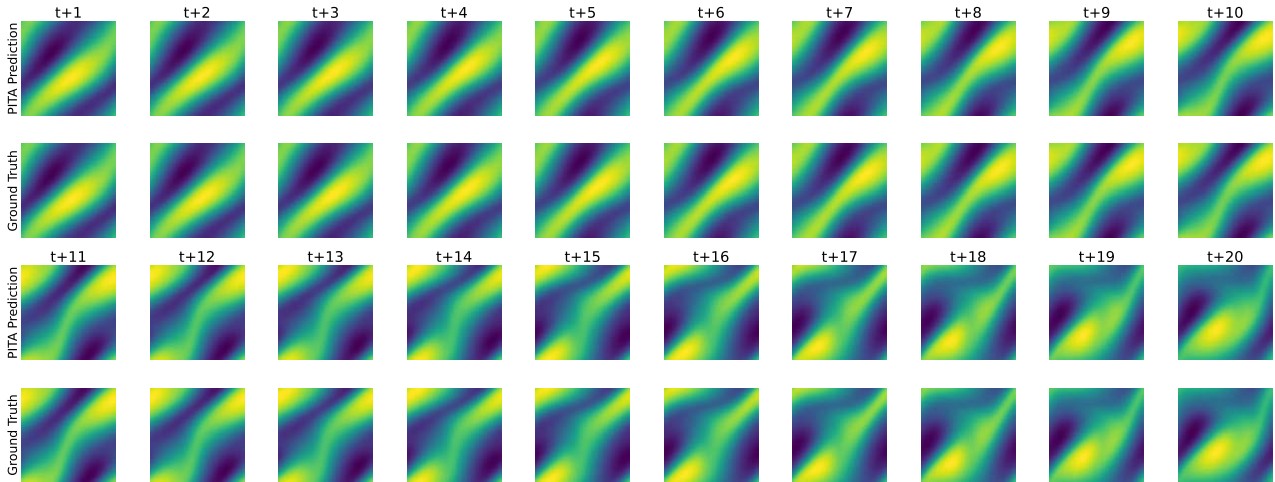

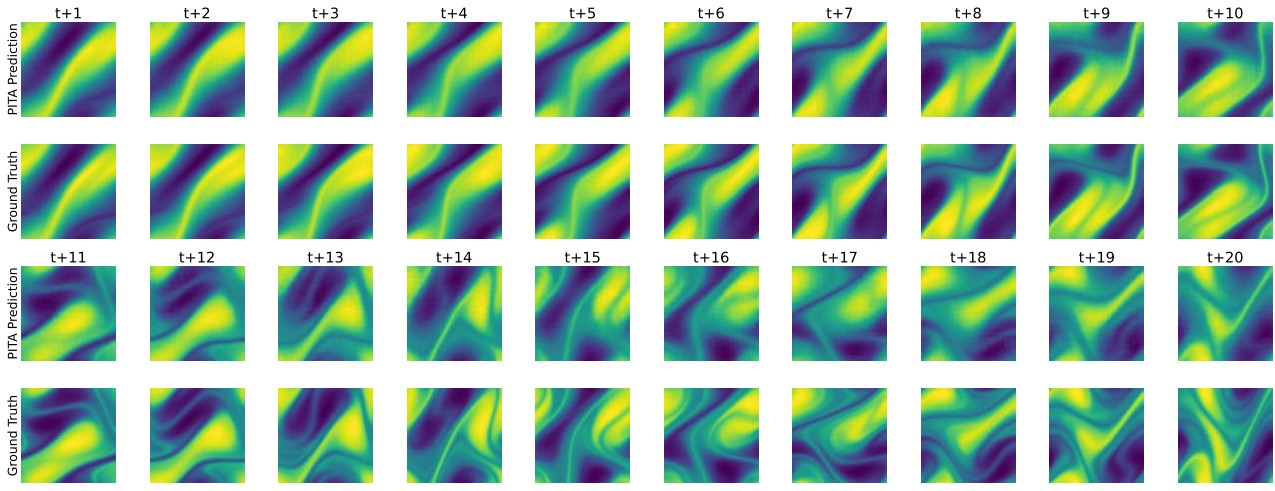

- $\nu = 1 \times 10^{-5}$

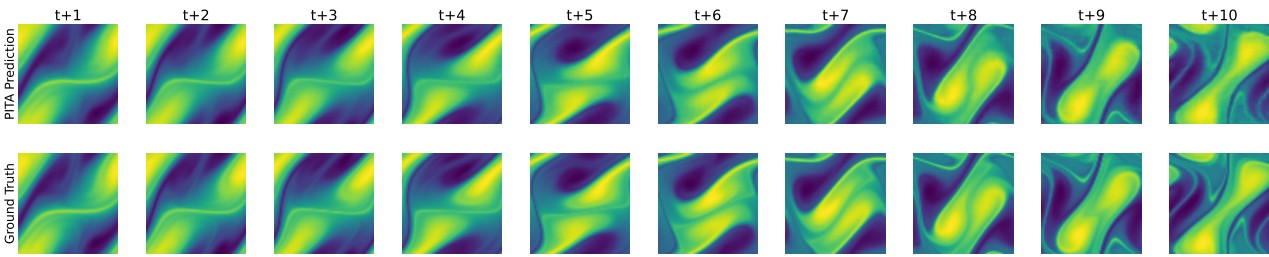

## F.3. PDEArena

Here we visulize the velocity $v$, pressure $p$, and density fields $\rho$ from two datasets from PDEArena-NS1/2: NS-Force and NS.

- NS-Force

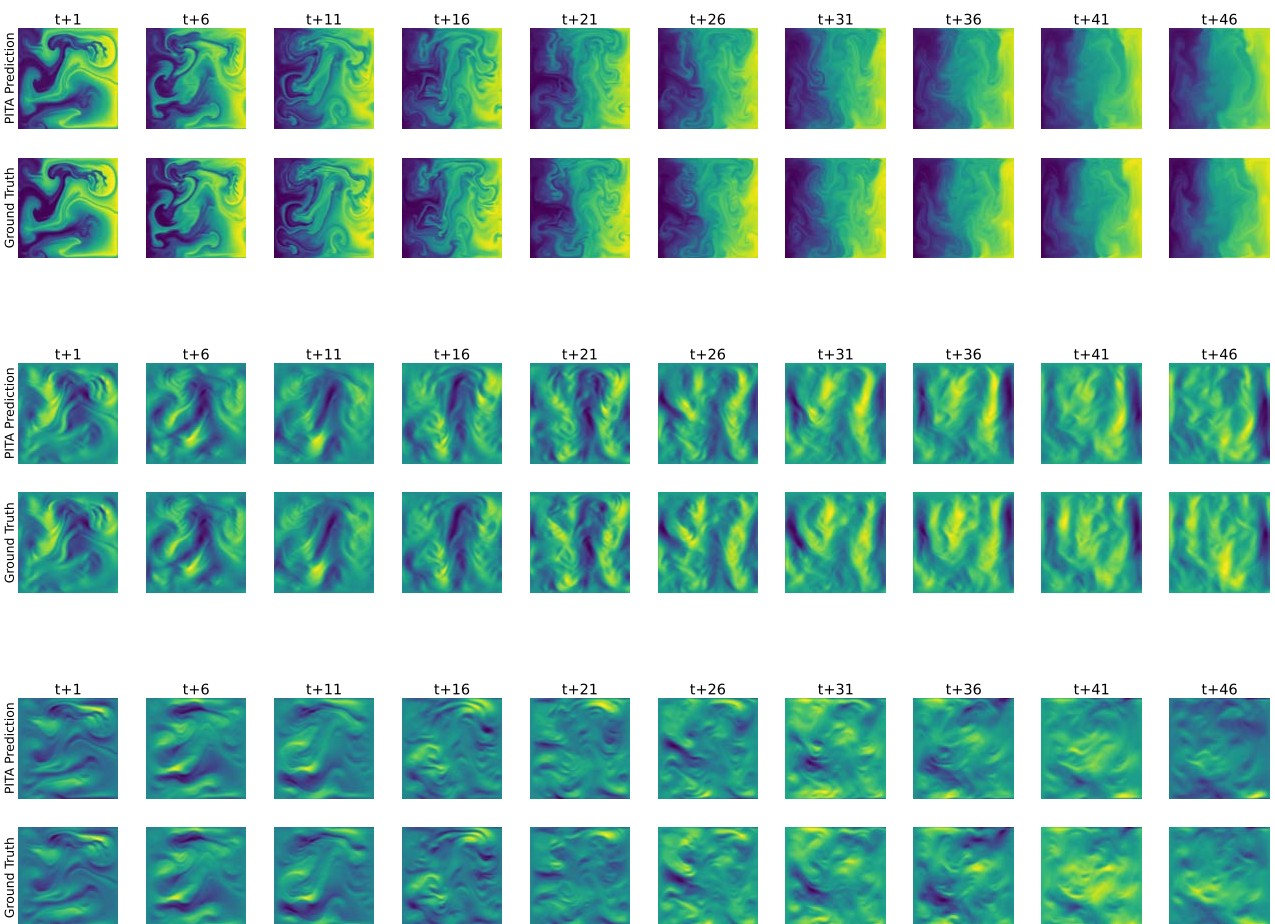

• NS

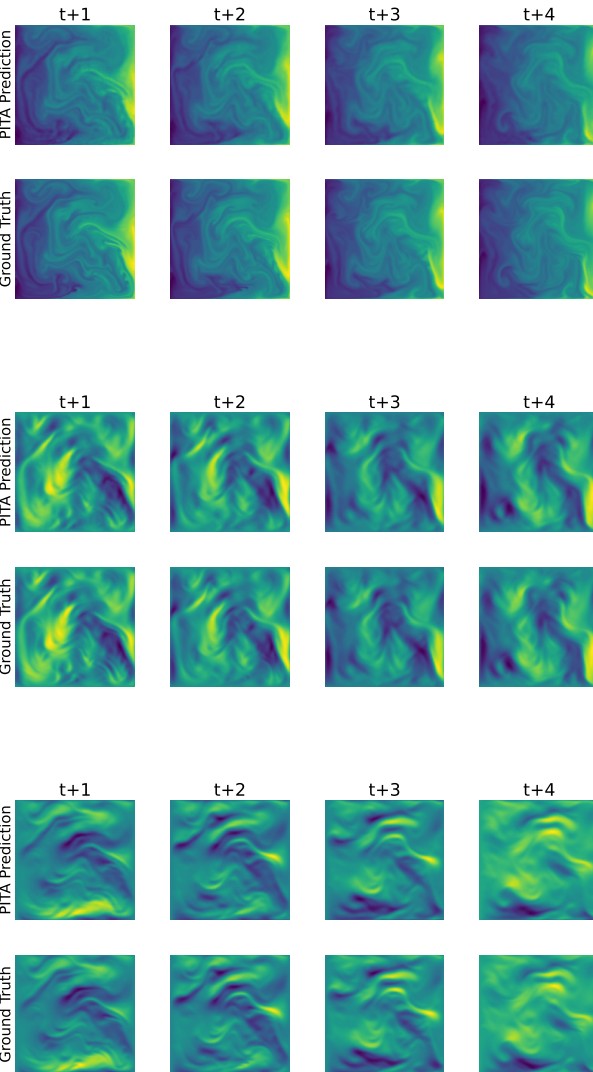

## F.4. PDEBench-CNS

Here we visualize the velocity fields $u$ and $v$, pressure $p$ and density $\rho$ of the compressible naive-stokes equation with different parameters.

- $(M, \eta) = (1, 0.01)$

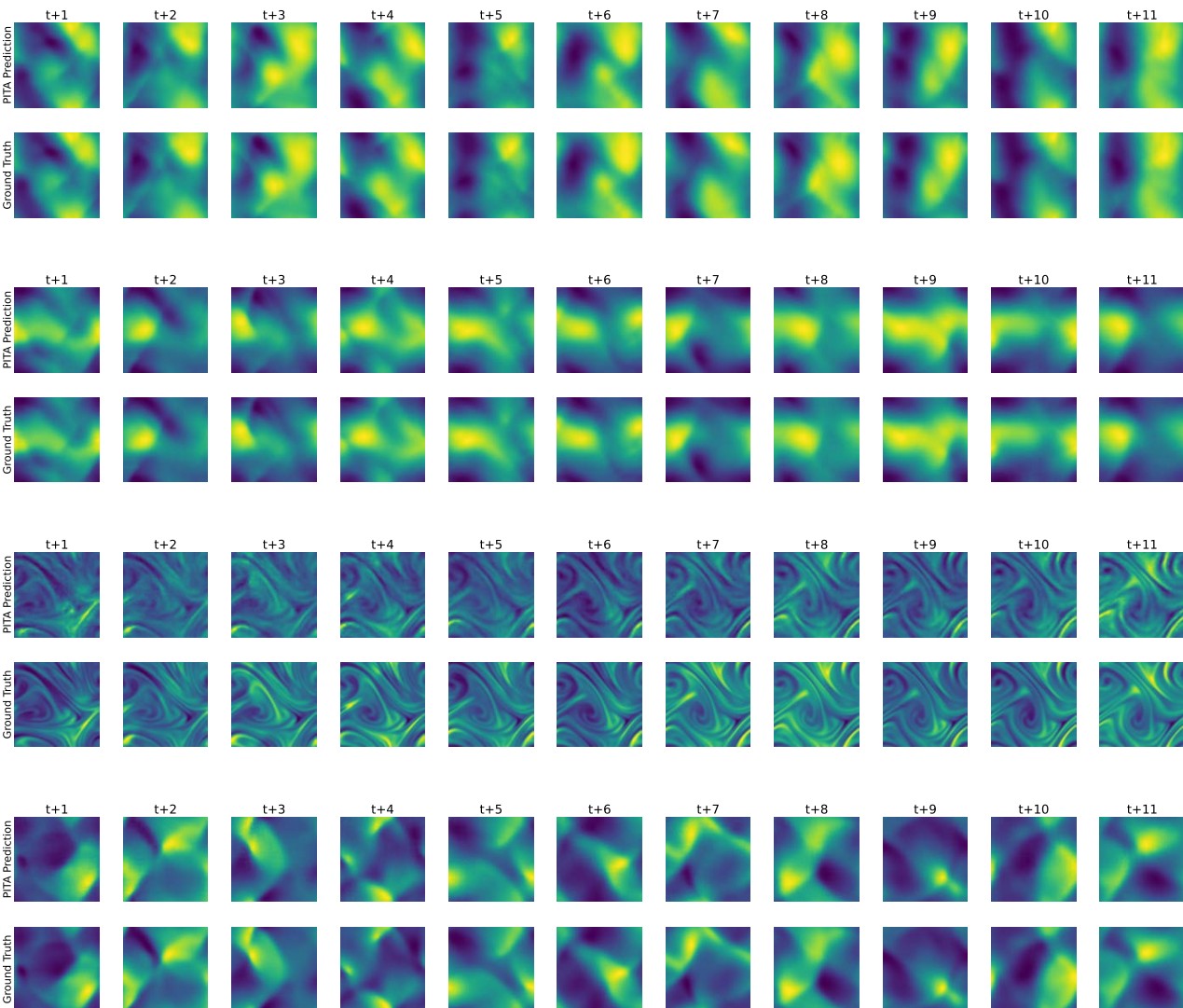

- $(M, \eta) = (0.1, 0.01)$

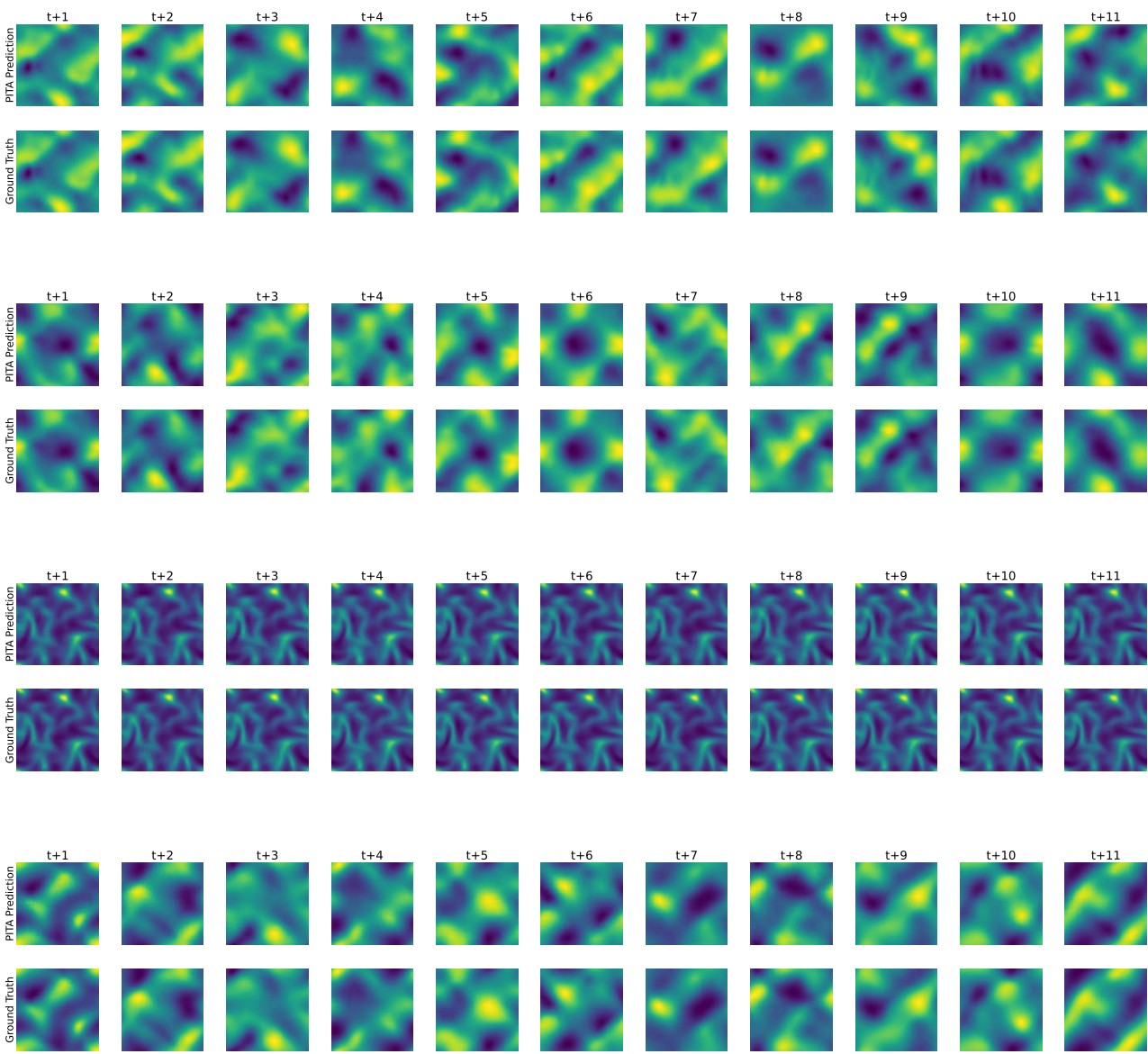

- $(M, \eta) = (1, 0.1)$

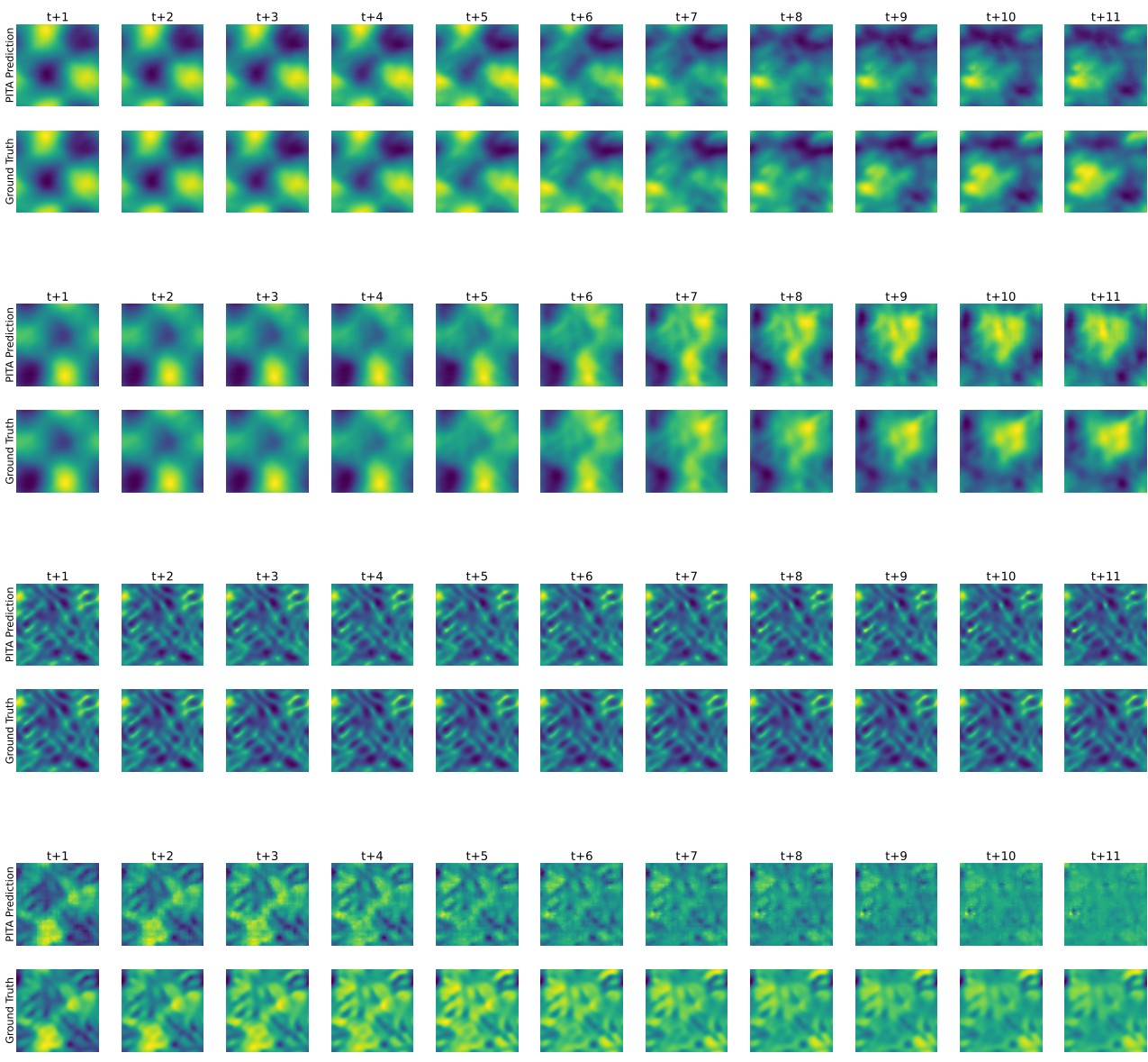

- $(M, \eta) = (0.1, 0.1)$

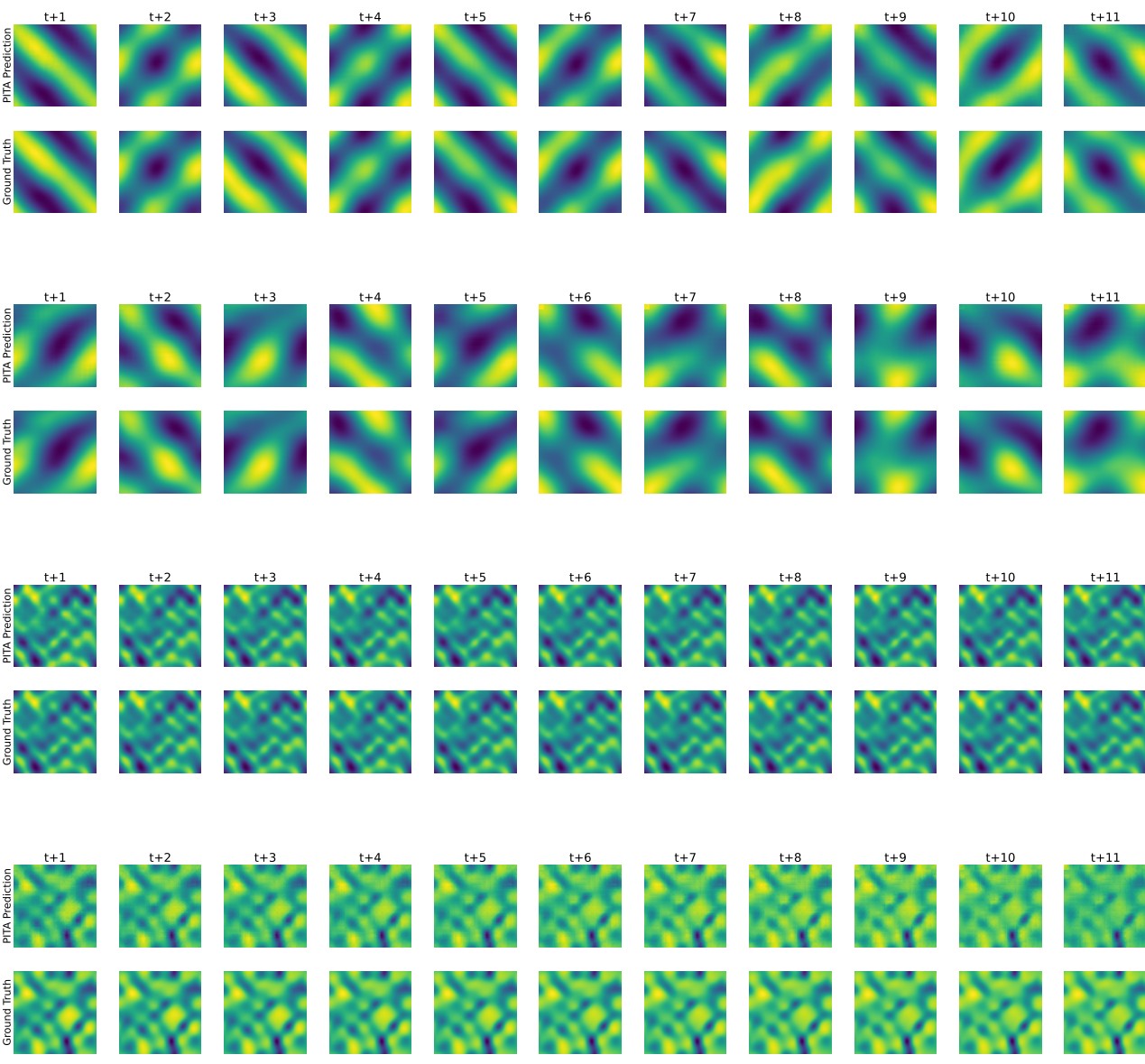

- $(M, \eta) = (1, 1 \times 10^{-8})$

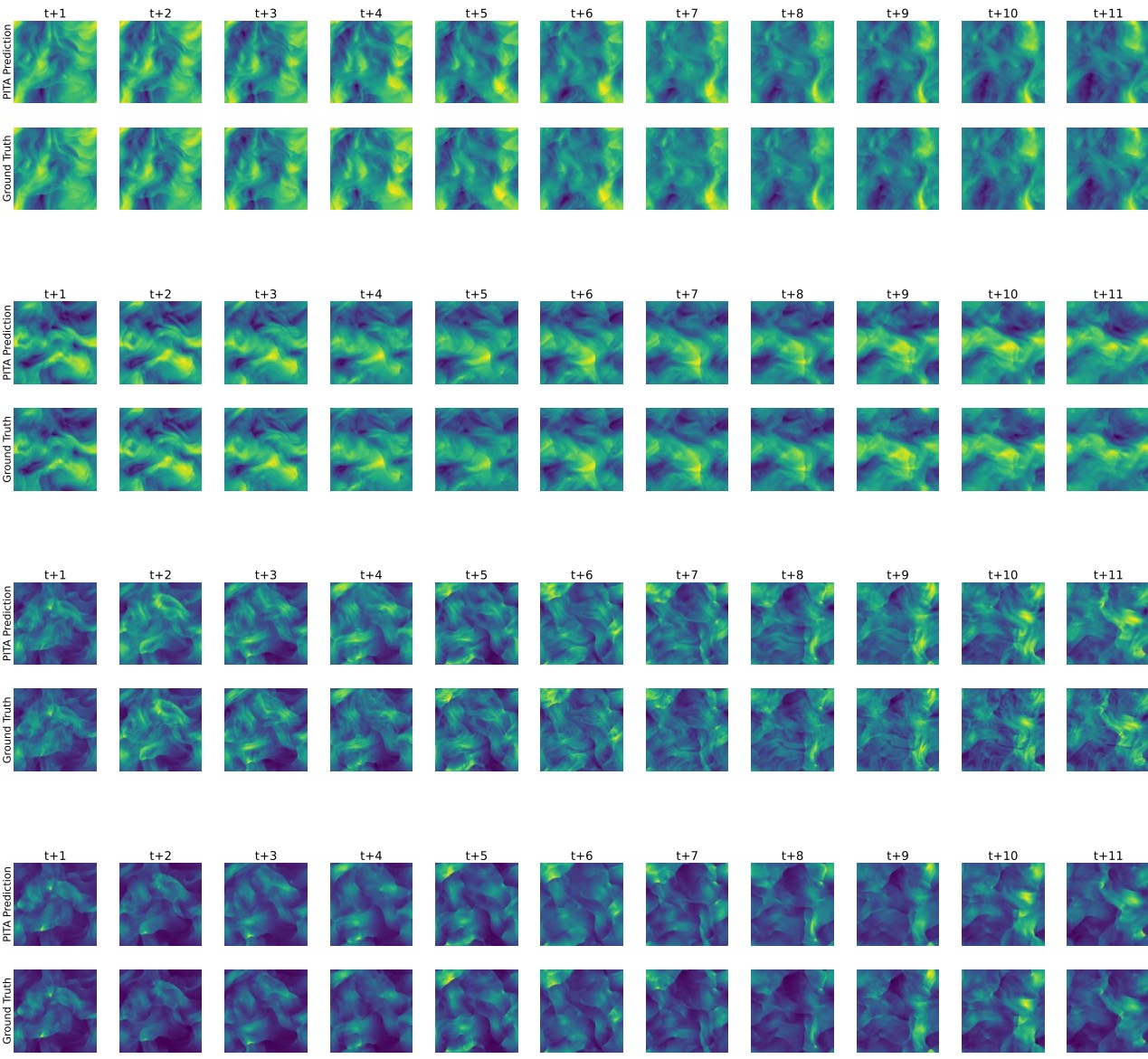

## F.5. Magnet-Viscous Burgers

Here we visualize the field $u$ from viscous burgers' equation.

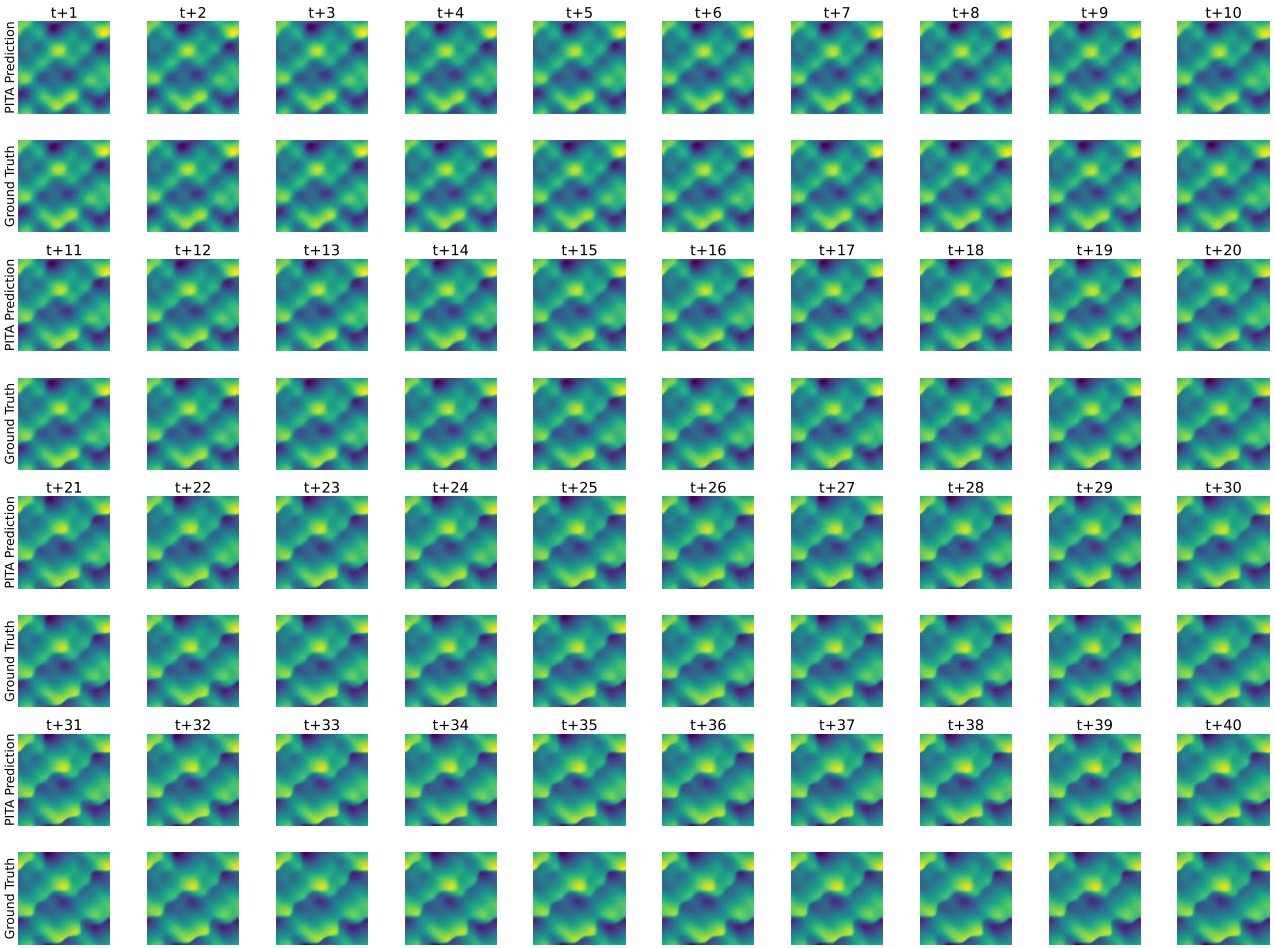

## F.6. CFDBench

Here we visualize the velocity and pressure fields $u$.

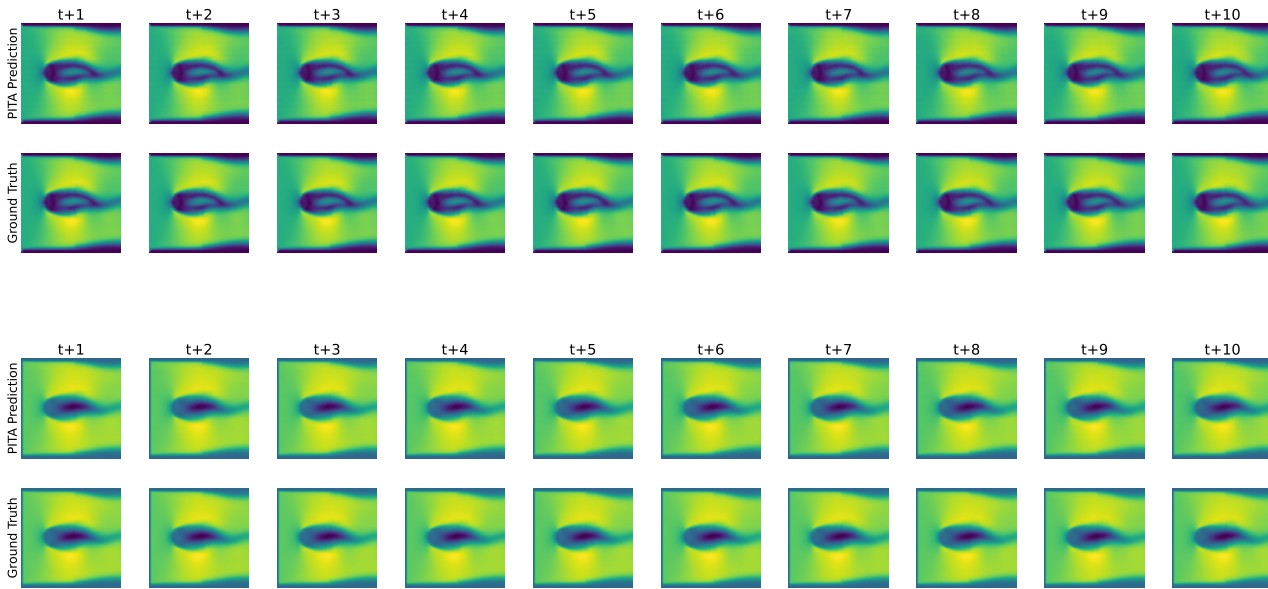

## F.7. PDEBench-SWE

Here we visualize water depth $h$ from the shallow-water equation.

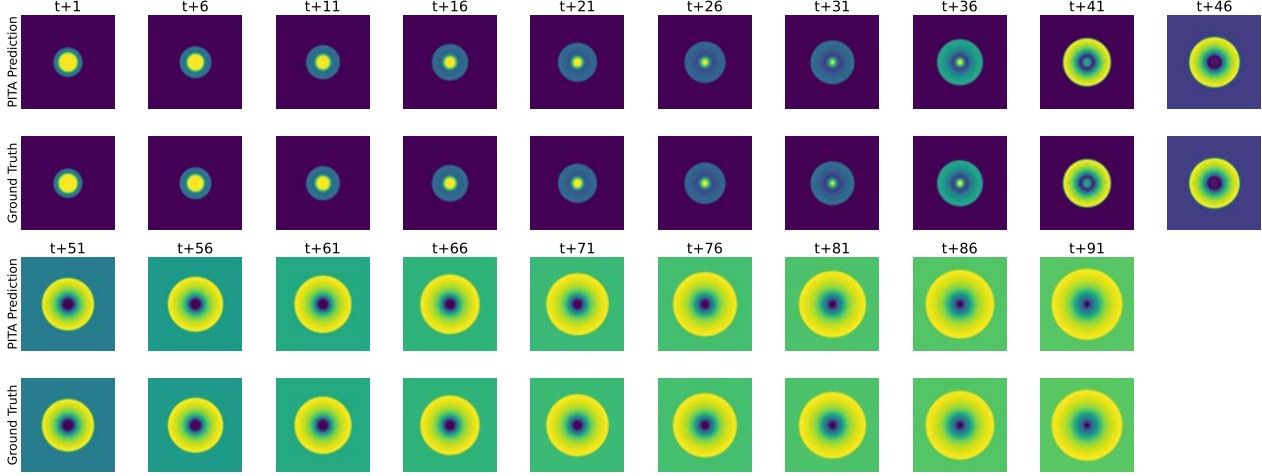

