# OpenReview forum: "Physics-informed Temporal Alignment for Auto-regressive PDE Foundation Models"
_ICML.cc/2025/Conference — ICML 2025 poster_

### Official Review · Reviewer_qJJU · 2025-03-09

**Overall Recommendation:** 3

**Summary:**

The authors studies autoregressive model which are used to make predictions about time series governed by a PDE system. To solve the shortcut problem for autoregressive models, the authors propose to use the training sequence to learn the governing equation, then train the autoregressive model so that they fit the training data and also the physical law learned.

**Claims And Evidence:**

All the results should include error bounds -- otherwise it is difficult to see whether the improvement is due to random chance or due to the actual methods.

The ablation studies seems to be okay, since it directly investigates the performance of each introduced components, however could also be tried for other tested dataset as well to make it more convincing.

I am also curious how important the PDE learning component is, and how robust PITA is to the component. Specifically, if the PDE regression component is not sufficient to model the true equation, how much would the learned dynamics suffer? Furthermore, it may be that not learning the PDE in full but rather providing a better PDE regularisation may also be enough (this is shown in Table 3 Task 1, where Lphy + Ldata does perform worse than PITA but not that much worse, and could potentially be improved to bypass directly learning the PDE in the first place too).

**Essential References Not Discussed:**

None as far as I am aware of.

**Experimental Designs Or Analyses:**

In terms of the shortcut problem that is mentioned by the authors -- while the results of PITA seems to be positive, it is unclear whether this is due to its success in solving the shortcut problem, or whether the loss term is able to simply better capture the physics bias in general. I'm not sure if the fact that long term trajectory accuracy is sufficient for this (there may be other reasons long-term trajectories perform poorer in other models).

**Methods And Evaluation Criteria:**

The benchmarks seem to be reasonable for physics-informed problems, however I'm not sure if more complex datasets (e.g. more high-dimensional or noisier real-life data) should also be used beyond those that are already typically studied in simpler PINNs and deep operator models. I do think that the fluid dynamics data might be trending towards these scenarios though and may be okay if these are also the norms used in existing benchmarks.

I notice that the authors have mentioned that the method requires more computation than other benchmarks. While the authors have given some numbers to this in the limitations section, they could be reported further as well for a clearer picture of the additional computational requirements for the methods.

**Other Comments Or Suggestions:**

None.

**Other Strengths And Weaknesses:**

No other points beyond what was addressed above.

**Questions For Authors:**

Most of the questions have been raised from above but will mention some of them here again:

1. Regarding the importance of the PDE solver in this case -- how reliant is the method on a adequately good design of the PDE discovery term?

2. Are the authors able to provide stronger proof that the loss term is able to reduce the shortcut problem, rather than just merely reducing the prediction error and attributing this to being due to the shortcut problem being solved?

**Relation To Broader Scientific Literature:**

This work seems to be relevant to learning physical systems with data and making predictions based on them. They therefore seem to be quite relevant to scientific applications where those types of PDEs are studied. Due to the equation discovery portion they could also be used for other types of time series observations as well, however this does not seem to be focused on in the current paper.

**Theoretical Claims:**

There are none in the paper.

---

> ### Author Rebuttal · Authors · 2025-04-01
>
> 1.Regarding the ablation studies on more datasets.
>
> Reply 1: We conducted additional experiments on three datasets that span different PDE domains and exhibit varied physical characteristics, with an average step length of 84. The results (https://anonymous.4open.science/r/PITA_1/Table2.pdf) consistently corroborate our main findings, further reinforcing the reliability of our conclusions.
>
> 2.Regarding the error bounds.
>
> Reply 2: Thanks for your insightful question. PITA is fundamentally a data-driven PDE-solving approach built upon auto-regressive baselines. To our knowledge, theoretical analysis of error bounds in data-driven PDE methods remains an open and challenging problem, which is currently beyond the scope of our work. Notably, baseline methods such as DPOT, FNO, and MPP also do not provide discussions on error bounds. Nonetheless, we fully agree with the reviewer’s point and will carefully consider error analysis in future.
>
> 3.Regarding the PDE learning component and the robustness of PITA.
> Reply 3: Even if equation discovery fails, our multi-task learning automatically down-weights the physics loss, causing the model to degrade into a dual-constraint mode governed by data loss and consistency loss. Here, the consistency loss specifically aligns macroscopic data patterns derived from the discovered dynamics. As shown in Table 3 of our submission, even without physics loss, PITA still surpasses the baseline by 43.02%. For longer trajectories, the consistency loss is also critical, as demonstrated in https://anonymous.4open.science/r/PITA_1/Table2.pdf.
>
> Additionally, we conducted a stress test by directly injecting noise into the physics-loss gradients, simulating severely incorrect physics guidance. Results in https://anonymous.4open.science/r/PITA_1/Table6.pdf and Table7.pdf show the robustness, and Table9.pdf indicates that PITA regrades as baseline performance under such attacks.
>
> Finally, while both PITA and PINN-based methods leverage physics constraints, PINNs require explicitly formulated governing equations (e.g., PINN-SR[1]). In contrast, PITA discovers the underlying PDE directly from data without physics priors, making PDE discovery not merely complementary but essential for learning unknown dynamics.
>
> 4.Regarding the evaluation on more complex dataset.
>
> Reply 4: We conducted experiments on synthetic 2D Kolmogorov turbulence flows—a more complex dataset with 150 temporal steps. Using DPOT-Ti model, our results (https://anonymous.4open.science/r/PITA_1/Table4.pdf) demonstrate that PITA substantially outperforms the baseline. Due to space constraints, please refer to the discussion in Reply 4 to Reviewer y7j5.
>
> 5.Regarding the additional computational cost.
>
> Reply 5: Thanks for your valuable suggestions. As shown in https://anonymous.4open.science/r/PITA_1/Train.pdf , the time consumption of the PDE discovery and alignment steps is constant and does not scale with the model size. Moreover, PITA performs PDE discovery and alignment only during the training phase, so its inference efficiency is identical to the baseline. Due to space limitations, please refer to the discussion in Reply 3 to Reviewer y7j5.
>
> 6.Regarding the shortcut problem and long-term trajectory accuracy.
>
> Reply 6: As discussed in Reply 4, PITA exhibits significant advantages in long-trajectory data. Following the accumulative error visualization in [1], we plotted the rolling-step MSE. Results (https://anonymous.4open.science/r/PITA_1/, see 'Shortcut' folder) show that the error remains stable and comparable to earlier time steps. Due to space constraints, please refer to the discussion in Reply 1 to Reviewer Xe1t.
>
> 7.Regarding other types of time series observations.
>
> Reply 7: We agree that the PITA framework can indeed be extended and adapted to other types of time series data. Your feedback has been incredibly insightful, and we are committed to incorporating this direction into future work.
>
> 8.Regarding the design of the PDE discovery term.
>
> Reply 8: We adopted the PDE-FIND library, which is widely utilized by PDE discovery methods such as SINDy, as a general and standard library in the field without requiring specialized design. As shown in https://anonymous.4open.science/r/PITA_1/Table3.pdf, experiments demonstrate that PITA remains effective even when more than half of the candidate terms in the library are removed randomly. This indicates PITA does not depend on carefully tailored terms.
>
> 9.Regarding the proof of loss term being able to reduce the shortcut problem.
>
> Reply 9: As discussed in Reply 3, 4, and 6, results provide strong evidence of PITA's promising potential for shortcut learning.
> We attribute this to two key components: (1) the physics loss, enforcing fine-grained alignment of physical dynamics, and (2) the consistency loss, aligning macroscopic data patterns over temporal windows, jointly mitigating error accumulation.
>
> [1] Physics-informed learning of governing equations from scarce data.

---

> > ### Comment · Reviewer_qJJU · 2025-04-06
> >
> > Thanks the reviewer for their response.
> >
> > I should clarify slightly -- in this sense, error bounds I am talking about the standard deviation in the reported empirical error values from the experiments. I understand that this area seems to lack more theoretical results so I wouldn't expect the generalisation error bounds to be reported anyway.
> >
> > I do think that the remaining points seem well-addressed though, so given the additional points already addressed I think it could be an accepted paper.

---

> > > ### Author Response · Authors · 2025-04-09
> > >
> > > We sincerely thank the reviewer for this helpful clarification and thoughtful feedback. In our previous response, we mistakenly interpreted “error bounds” as referring exclusively to theoretical error bound analysis. Your clarification regarding empirical error analysis (e.g., standard deviation) greatly enhances our understanding of the reviewer perspective, highlighting the practical importance of empirical measures in assessing model reliability [1,2].
> > >
> > > Due to the imminent rebuttal deadline (April 8th) and resource constraints, conducting comprehensive experiments across all datasets (12 datasets), models (MPP, DPOT, FNO), and scales (Ti, S, M, L) was not feasible. Therefore, we selected two representative out-of-distribution datasets—the Burgers (50 Steps) and PDE-Bench CNS-(M, η) (20 Steps)—as detailed in Section 4.2. These datasets were deliberately chosen because they were not included in the pre-training phase, thus providing a rigorous evaluation of model reliability. We compared PITA against two baseline models, DPOT and FNO, across the Ti, S, and M scales. Each experiment was repeated five times with different random seeds, and full results can be found at https://anonymous.4open.science/r/PITA_1/Table11.pdf, https://anonymous.4open.science/r/PITA_1/Table12.pdf, https://anonymous.4open.science/r/PITA_1/std_burgers.pdf and https://anonymous.4open.science/r/PITA_1/std_ns2d.pdf.
> > >
> > > Our selection was motivated by the following considerations:
> > >
> > > (1) Dataset OOD nature: The chosen datasets differ significantly from the training distribution. Specifically, the Burgers equation represents a PDE family distinct from the training data, thereby posing a strong generalization challenge [3].
> > >
> > > (2) Dynamics length variability: Trajectories of varying lengths test the stability and reliability of models in capturing both short-term and long-term dynamics.
> > >
> > > (3) Model scale analysis: Evaluating multiple model scales allows us to examine variance patterns and reliability characteristics dependent on model complexity.
> > >
> > > As shown in https://anonymous.4open.science/r/PITA_1/std_burgers.pdf and https://anonymous.4open.science/r/PITA_1/std_ns2d.pdf, our empirical results demonstrate that PITA hardly increases the baseline’s error bars. In fact, in most cases, it simultaneously reduces both prediction errors and associated error bars. Moreover, when evaluating longer trajectories (see https://anonymous.4open.science/r/PITA_1/std_burgers.pdf), PITA consistently achieves notably lower standard deviations compared to auto-regressive approaches on models like DPOT-Ti, DPOT-M and FNO-M. These tight standard deviations strongly indicate that PITA's improvements are statistically significant and arise inherently from its architecture, rather than from random stochastic variations.
> > >
> > > Together with our previous stability analyses (see Replies 3, 4, and 8), these results further validate the empirical reliability and generalization capabilities of PITA. Nonetheless, we acknowledge that this empirical analysis remains preliminary due to deadline and resource limitations. We are fully committed to expanding and deepening these experiments and analyses in future revisions.
> > >
> > > Finally, we sincerely appreciate your insightful suggestions, which have significantly improved the clarity of our work and helped better position PITA within the autoregressive modeling landscape. We also appreciate your positive assessment of our prior responses and remain committed to implementing all suggested revisions. Should you find it appropriate, we would be grateful if you could consider updating 'Overall Recommendation' in the system, reflecting your final decision to the Area Chair.
> > >
> > >
> > > [1]Lippe P, Veeling B, Perdikaris P, et al. Pde-refiner: Achieving accurate long rollouts with neural PDE solvers[J]. Advances in Neural Information Processing Systems, 2023, 36: 67398-67433.
> > >
> > > [2]Bar-Sinai Y, Hoyer S, Hickey J, et al. Learning data-driven discretizations for partial differential equations[J]. Proceedings of the National Academy of Sciences, 2019, 116(31): 15344-15349.
> > >
> > > [3]Basto M, Semiao V, Calheiros F. Dynamics in spectral solutions of Burgers equation[J]. Journal of Computational and Applied Mathematics, 2007, 205(1): 296-304.

---

### Official Review · Reviewer_y7j5 · 2025-03-09

**Overall Recommendation:** 3

**Summary:**

The paper *"Physics-informed Temporal Alignment for Auto-regressive PDE Foundation Models"* introduces **Physics-Informed Temporal Alignment (PITA)**, a self-supervised learning framework designed to address the error accumulation problem in auto-regressive PDE foundation models. Instead of relying on predefined physics priors, PITA **discovers governing equations from observation data** and integrates them into the model’s training process through a combination of **data loss, physics loss, and consistency loss**. By aligning the physical dynamics at different time steps, PITA ensures that predictions remain consistent with underlying physical laws, reducing shortcut learning and improving generalization to out-of-distribution data. Extensive experiments across **12 datasets** show that PITA significantly enhances both short and long-term prediction accuracy, outperforming existing auto-regressive methods while maintaining scalability.

**Claims And Evidence:**

Yes. I do not see significant problems in this paper.

**Essential References Not Discussed:**

OmniArch: Building the Foundation Model for Scientific Computing

Authors: Tianyu Chen, Haoyi Zhou, Ying Li, et al.​
Venue: arXiv preprint arXiv:2402.16014, 2024.​
Relevance: OmniArch proposes a unified architecture to tackle multi-scale and multi-physics scientific computing challenges by integrating physical alignment into the modeling process. This approach aligns with PITA's goal of incorporating physics-based constraints to improve model predictions. ​


PINNsFormer: A Transformer-Based Framework for Physics-Informed Neural Networks

Authors: Zhiyuan Zhao, Xueying Ding, B. Aditya Prakash​
Venue: arXiv preprint arXiv:2307.11833, 2023.​
Relevance: PINNsFormer introduces a framework that leverages transformer architectures to capture temporal dependencies in physics-informed neural networks. This methodology is pertinent to PITA's focus on aligning model predictions with temporal dynamics governed by PDEs. ​


PIMRL: Physics-Informed Multi-Scale Recurrent Learning for Spatiotemporal Prediction

Authors: Wan Han, Qi Wang, Hao Sun​
Venue: International Conference on Learning Representations (ICLR), 2025.​
Relevance: PIMRL introduces a multi-scale learning framework that effectively leverages multi-scale data for spatiotemporal dynamics prediction, addressing challenges similar to those targeted by PITA. ​

**Experimental Designs Or Analyses:**

Generally, the experiments are comprehensive.

However, I have additional two questions:

1. Long-Horizon Stability – There is no explicit analysis of how PITA performs in very long rollouts (e.g., 100+ steps), particularly for chaotic PDEs, where error accumulation could still occur.

2. Computational Overhead – The training time, inference speed, and memory usage of PITA are not reported, making it unclear how computationally expensive the PDE discovery process is compared to baseline models.

**Methods And Evaluation Criteria:**

Overall, the proposed methodology and evaluation criteria are well-justified, relevant, and appropriate for assessing the challenges associated with auto-regressive PDE modeling.

However,  the governing equation discovery process introduces an additional computational step after each auto-regressive prediction. While this ensures physics-informed corrections, it may significantly increase training time, particularly for large-scale PDE simulations. The paper does not quantify the computational cost trade-off between accuracy improvements and increased training time. There is a need for benchmarks on training efficiency, memory consumption, and scalability across different model sizes.

**Other Comments Or Suggestions:**

Please see above reviews.

**Other Strengths And Weaknesses:**

Please see above reviews.

**Questions For Authors:**

Could you provide detailed metrics on the computational cost associated with the governing equation discovery process within PITA, including training time, memory usage, and scalability across different model sizes?

How does PITA perform when the underlying PDEs exhibit complex, high-dimensional, or non-sparse structures? Have you tested the framework on systems where the governing equations are highly nonlinear or do not conform to a sparse representation?

**Relation To Broader Scientific Literature:**

[1] This approach aligns with the broader scientific literature on Physics-Informed Neural Networks (PINNs), which embed physical laws directly into neural network training to solve both forward and inverse problems involving PDEs. PINNs leverage governing equations to guide the learning process, ensuring that neural network solutions adhere to known physical principles. ​

[2] The development of Neural Operators represents a significant advancement in this domain. Neural Operators are designed to learn mappings between infinite-dimensional function spaces, enabling the modeling of complex physical systems with improved efficiency and accuracy. They have been applied to various scientific and engineering disciplines, including turbulent flow modeling and computational mechanics, demonstrating their capability to handle complex PDE solutions across varying input conditions and geometries.​

[3] The Theory of Functional Connections (TFC) has been applied in neural networks to enhance the performance of PINNs by effectively eliminating constraints from the optimization process. This integration significantly improves computational efficiency and accuracy, enabling the resolution of complex problems with greater ease.

**Theoretical Claims:**

No theoretical claims.

---

> ### Author Rebuttal · Authors · 2025-04-01
>
> 1.Regarding Long-Horizon Stability
>
> Reply 1：To rigorously evaluate PITA’s capacity for modeling chaotic dynamical systems under extended temporal extrapolation, we conducted controlled experiments on synthetic 2D Kolmogorov turbulence flows, which is a canonical benchmark for chaotic PDE systems exhibiting multiscale energy cascades and nonlinear dissipation processes. Our turbulence simulation dataset contains 300 spatiotemporal trajectories with 64×64 grid resolution and 150 temporal steps, intentionally designed to challenge long-term forecasting fidelity. Both methodologies leverage the same pretrained DPOT-Ti model. As shown in https://anonymous.4open.science/r/PITA_1/Table4.pdf, our results demonstrate that PITA significantly outperforms the auto-regressive baseline in capturing the intricate patterns of chaotic fluid evolution. Specifically, PITA exhibits superior alignment with the underlying physical dynamics, showcasing its enhanced ability to model complex, long-term behaviors in turbulent systems.
>
> 2.Regarding Essential References.
>
> Reply2: Thanks for your valuable suggestion. We apologize for having cited an outdated version of OmniArch as:
> Building flexible machine learning models for scientific computing at scale.We will ensure that the recommended references are properly discussed in the main text of the revised manuscript.
>
> 3.Regarding the computational cost.
>
> Reply 3: We appreciate the reviewer’s concern regarding computational cost. The additional overhead from PDE discovery and temporal alignment primarily arises from three steps detailed in Section 3.3: building candidate libraries, sparse regression, and alternating direction optimization. As shown in https://anonymous.4open.science/r/PITA_1/Train.pdf, the time overhead of these steps remains constant and does not scale with model size.
> Regarding inference efficiency, since PDE discovery and alignment occur only during training, PITA's inference efficiency matches that of the baseline, while consistently outperforming it on long-term predictions (Figure 4, Appendix D.2).
> In terms of memory usage, PITA incurs no additional GPU memory overhead compared to auto-regressive methods. It does require moderate extra CPU memory for building candidate libraries and sparse regression (https://anonymous.4open.science/r/PITA_1/CPU.pdf). Importantly, this CPU memory overhead is fixed.
> Collectively, these features render PITA fundamentally scalable: its computational overhead is decoupled from model dimensionality, effectively overcoming traditional memory limitations in physics-informed methods. This architecture-agnostic scalability supports seamless adaptation to billion-parameter models with bounded resource consumption. For instance, a 500M-parameter L-level model incurs only 0.517 additional seconds per batch (~20% increase), yet achieves a notable 31.61% performance improvement on long-trajectory prediction (https://anonymous.4open.science/r/PITA_1/Table1.pdf).
>
> 4.Regarding the performance of PITA tested on underlying PDEs that exhibit (1) complex, high-dimensional, or non-sparse structures; and (2) high nonlinearity or structures that do not conform to a sparse representation.
>
> Reply 4: The experimental validation of the PITA framework is conducted on a series of high-dimensional PDE datasets spanning diverse physical systems, including fluid dynamics, reaction-diffusion processes, and nonlinear wave phenomena, which are summarized as follows:
> a.FNO-NS-ν and PDEArena Datasets
> Derived from the 2D incompressible Navier-Stokes equations, these datasets involve strongly coupled velocity-pressure fields and nonlinear advection terms, posing challenges in modeling multiscale interactions. The time-varying external force $f(x,t)$ in PDE-Arena-NS-Force further introduces non-stationary dynamics, requiring adaptive modeling capabilities. PITA achieves a 21.82% improvement in prediction accuracy compared to autoregressive baselines.
> b.PDEBench-DR
> The reaction terms $R_u(u,v)$ and $R_v(u,v)$ induce chaotic dynamics and Turing patterns, reflecting the complexity of high-order nonlinear systems. PITA demonstrates a 15.48% performance gain in resolving multistable patterns and chaotic trajectories.
> c.Viscous Burgers Equation
> Characterized by shock formation and nonlinear advection-dominated regimes, this dataset validates robustness in handling discontinuous solutions. PITA achieves a 19.41% error reduction in shock-capturing tasks compared to traditional solvers.
> d.As discussed in Reply 1 regarding the Kolmogorov Turbulence Flow, it is a canonical benchmark for chaotic PDE systems exhibiting multiscale energy cascades and nonlinear dissipation processes. As shown in https://anonymous.4open.science/r/PITA_1/Table4.pdf, our results demonstrate that PITA significantly outperforms the auto-regressive baseline in capturing the intricate patterns of chaotic fluid evolution.

---

> > ### Comment · Reviewer_y7j5 · 2025-04-04
> >
> > I have read the rebuttal. I decide to keep my rate.

---

> > > ### Author Response · Authors · 2025-04-05
> > >
> > > Thank you for your thoughtful evaluation and the positive comments about our work. We truly appreciate the time and effort you have devoted to providing constructive feedback, which has been instrumental in helping us refine our paper. In our previous response, we have made every effort to address all the issues raised, and we remain fully committed to clarifying any remaining concerns you may have.
> > >
> > > We deeply value your expertise and the critical role you play in upholding academic standards. If any part of our response remains unclear or if further discussion would be helpful, we would be more than willing to engage further, reflecting our sincere commitment to the quality and clarity of our research.

---

### Official Review · Reviewer_Xe1t · 2025-03-11

**Overall Recommendation:** 4

**Summary:**

This paper focuses on the shortcut bias that can occur in PDE auto regressive models. The proposed method suggests enhanced predictions using some physical knowledge extracted from data, using sparse regression. The results show some improvement in the rollout prediction.

## update after rebuttal
After the rebuttal discussion, I am in favor of acceptance of the paper.

**Claims And Evidence:**

The paper shows extensive experiment on the gain in term of performances of the proposed method. However, I think that some experiment highlighting the shortcut problem would benefit to the argumentation. While it is detailed as being the main focus of the paper, it is hard for the reader to identify whether the proposed method succeeds to solve this issue. Some visualizations are proposed in the appendix, but maybe some focused on these could be a first illustration? It is hard to identify from mse rollout if the proposed method effectively handles this problem or not.

**Essential References Not Discussed:**

NA up to my knowledge

**Experimental Designs Or Analyses:**

Experiments highlight the improvement in performances when using PITA. However, no comparison on the shortcut problem is provided, which could help in assessing the effectiveness and understanding the effect of the proposed method on the predicted trajectories. Moreover, since the method involves solving sparse regression, I think providing training and inference time comparison is an important point.

**Methods And Evaluation Criteria:**

The proposed method is evaluated on several standard and recent models and datasets of the literature.

**Other Comments Or Suggestions:**

- Line 149 right column, w not introduced before
- Lines 163, 164, theta not introduced, are they the parameters of the models being used for the prediction task? Ie DPOT, MPP…? (are they the same parameters as line 244 right column?)

**Other Strengths And Weaknesses:**

- I appreciate the limitation section, providing insights on the limitations of the methods. While reading the paper, the 2 main limitations mentioned were the one I thought about. However, I think this would be interesting to illustrate these limitations with some ablation in the appendices.
- No computational time comparison is provided, while it is a great problem for PDE solving (it is discussed in the limitations section). Moreover, adding regression suggest an increase of the computational time. This should be detailed in the appendix.
- As discussed in the limitation section, It would complement the argumentation to evaluate the performances of the PITA method with uncomplete library terms, to evaluate the sensibility of the method to the prior assumption made as well as wrt the library size.
- The paper is well written, easy to follow and shows extensive experiments, well detailed.

**Questions For Authors:**

- 1/ It is well known in the literature that adding physical losses in the training objective can lead to unstable or long optimization. How does behaves PITA with respect to this? Does it complicate training?
- 2/ How behaves PITA when removing the data loss? Are L_con and L_Phy sufficient?
- 3/ From what I understood, the sparse regression is computed on downsampled trajectories?
- 4/ How are built $\lambda^*$? What happens if the true PDE coefficients are taken from an incomplete library functions?

**Relation To Broader Scientific Literature:**

Maybe that providing an illustration of what the shortcut problem is and more importantly, what does it implies for PDE would give insights to the reader (using standard methods of the literature).

**Theoretical Claims:**

NA: no theoretical claims in the paper.

---

> ### Author Rebuttal · Authors · 2025-04-01
>
> 1.Regarding the comparison of the shortcut problem.
>
> Reply 1: Thanks for your insightful comments. We followed the error visualization methodology in [1], plotting rolling-step MSE for each long-term dataset, as shown in the Shortcut folder of https://anonymous.4open.science/r/PITA_1/ (see 'Shortcut' folder). The results show that on datasets such as PDEBench-SWE, FNO-NS-1e-3, and FNO-NS-1e-4, the prediction error does not exhibit significant accumulation as time progresses. While errors exhibit a slight increasing trend over time in other datasets, PITA still reduces error accumulation compared to the auto-regressive method. This observation strongly indicates that PITA effectively mitigates error propagation over extended temporal sequences.
>
> 2.Regarding additional computational cost.
>
> Reply 2: Due to space limitations, please refer to the discussion of this topic in Reply 3 to Reviewer y7j5. We kindly invite you to consult that section for a detailed discussion.
>
> 3.Regarding illustrating the shortcut.
>
> Reply 3: Thanks for your suggestion. In the revision, we will incorporate a detailed explanation of the shortcut problem, including its definition, standard approaches, and its specific implications for PDE research.
>
> 4.Regarding the performance of the PITA with incomplete library terms.
>
> Reply 4: To evaluate the impact of the incomplete library, we conducted experiments with subsampled candidate term sets, as shown in https://anonymous.4open.science/r/PITA_1/Table3.pdf. When randomly retaining 50% of the library terms, the test loss on PDEBench-SWE increases from a baseline of 0.00137 (achieved with full-term libraries) to 0.00213. Notably, this performance degradation diminishes when preserving 80% of library terms, where the test loss stabilizes at 0.00135 – statistically comparable to the full-library configuration.  Crucially, our findings indicate that the library architecture requires no specialized customization for individual PDE systems.
>
> 5.Regarding Line 149, right column.
>
> Reply 5: We sincerely apologize for the oversight. The parameter w specifically denotes the resolution of the downsampled grid.
>
> 6.Regarding Lines 163, 164.
>
> Reply 6: Thanks for your meticulous attention to conceptual consistency. The notation θ in lines 163–164 indeed refers to the trainable parameters of models (e.g., FNO, DPOT, MPP), which aligns precisely with the parameter definition in Line 244.
>
> 7.Regarding the stability of physical losses in the training objective.
>
> Reply 7: Prior studies highlight a trade-off between data loss and physics loss in multi-task optimization, caused by conflicting gradient directions, which can result in unstable training or slow convergence [3,6].
> Inspired by [2], we adopt an uncertainty-based weighting strategy to balance the three losses, as detailed in Sec. 3.4. As shown in Table 3, PITA achieves a 48.97% improvement in test accuracy compared to the auto-regressive baseline. Furthermore, compared to manually specified weights for different losses (Table 3, Task 4), the uncertainty-based weighting strategy shows significant advantages.
>
> 8.Regarding removing the data loss.
>
> Reply 8: PITA fundamentally relies on autoregressive, data-driven signals. Removing data loss disrupts its ability to capture intrinsic PDE dynamics. Abandoning the data-driven paradigm entirely would require reconstructing the foundation model architecture, like Spline-PINN, or stronger physical priors, potentially limiting generalizability to unknown dynamics.
>
> 9.Regarding the computation of the sparse regression.
>
> Reply 9: Sparse regression indeed performs on temporally and spatially downsampled trajectories, as systematically validated in Table 3 of the submission. Temporal downsampling and spatial subsampling force the model to capture essential system dynamics and scale-invariant patterns while reducing noise-induced overfitting, which aligns with findings in PDE-Find [5].
>
> 10.Regarding 𝜆∗ and incomplete library functions.
>
> Reply 10: Both $\lambda^*$ and $\lambda_i(\theta)$ are computed via sparse regression algorithms, as detailed in Algorithm 1, with the distinction that $\lambda^*$is derived from ground-truth data while $\lambda_i(\theta)$originates from predicted data.
> Moreover, we kindly refer the reviewer to Reply 4 for a detailed discussion on the issue of incomplete candidate libraries.
>
> [1] A WENO-based method of lines transpose approach for Vlasov simulations.
>
> [2] Multi-task learning using uncertainty to weigh losses for scene geometry and semantics.
>
> [3] Multi-task learning as multi-objective optimization.
>
> [4] Spline-pinn: Approaching pdes without data using fast, physics-informed hermite-spline cnns.
>
> [5] Data-driven discovery of partial differential equations.
>
> [6] PINN training using biobjective optimization: The trade-off between data loss and residual loss.

---

> > ### Comment · Reviewer_Xe1t · 2025-04-03
> >
> > I thank authors for their answers to my concerns and the additional experiments. Below I ask additional questions based on the rebuttal answers.
> >
> > 1/ Could you comment on the behavior on SWE and dr_pdb. Why does the error decrease at the beginning of the trajectory? Why this oscillatory behavior? Other figures are convincing.
> >
> > 4/ PITA seems to remain consistent when removing some terms of the library. Have you checked when removing important terms? For example, How does the model behaves when the term of the basis are not relevant for the regression? I believe that some important terms are still in the 50% kept?
> >
> > With the suggested revision (2, 3, 5, 6 and additional ablations), I will vote for acceptance.

---

> > > ### Author Response · Authors · 2025-04-05
> > >
> > > Response to 1/:
> > >
> > > Thanks for your insightful comments. We observed that compared to other datasets, SWE and dr_pdb data exhibit slight dynamic evolution in the early stage, especially in SWE, where the dynamics show an oscillatory trend.
> > > As shown in https://anonymous.4open.science/r/PITA_1/SSIM_SWE.pdf, we present the data SSIM (structural similarity) between consecutive time steps for the SWE dataset. **The numerical evolution of the dynamics is subtle and exhibits random behavior, as evidenced by the high (0.9975–0.9945) and fluctuating SSIM values, indicating that the dynamic evolution is not significant. Under auto-regressive prediction, even simply copying the value from the previous time step as the prediction for the next can result in relatively low errors (MSE). Therefore, we speculate that the findings above contribute to the oscillatory behavior of the MSE observed in the early trajectory**.
> > >
> > > Further considering both https://anonymous.4open.science/r/PITA_1/SSIM_SWE.pdf and https://anonymous.4open.science/r/PITA_1/Shortcut/mse_comparison_SWE.pdf, **we can see that in later stages, when the SSIM for SWE decreases, the auto-regressive method exhibits noticeable error accumulation as the dynamics become more turbulent. **
> > >
> > > In contrast, in the FNO1e04 dataset, we observe a significant and steady decrease in SSIM values (0.96-0.82), with a rapid decline compared to SWE, as shown in https://anonymous.4open.science/r/PITA_1/SSIM_FNO1e04.pdf. **This indicates a drastic dynamic evolution, leading to increased prediction errors. If the model simply replicates the previous data pattern, it results in pronounced error accumulation. The SSIM trend is associated with this error accumulation in the auto-regressive method, as shown in https://anonymous.4open.science/r/PITA_1/Shortcut/mse_comparison_FNO1e04.pdf. ** PITA, however, effectively suppresses error accumulation in both the SWE and FNO1e04 datasets, addressing the limitations of the existing auto-regressive models.
> > >
> > > Response to 4/:
> > >
> > > Thanks for your insightful discussion. We followed the reviewer’s suggestion and removed important terms to observe PITA's performance. This experiment was conducted on the Burgers dataset, which was not part of the pre-training process, thereby providing a clearer insight into how the terms affect PITA's performance. The Burgers’ equation is composed of a time-dependent term, $\partial_t \boldsymbol{u}$, a diffusion term $\beta \Delta \boldsymbol{u}$ and a convective term $\boldsymbol{u}\nabla \boldsymbol{u}$. These terms are indispensable for capturing the underlying dynamics of the system [1].
> > >
> > > As shown in https://anonymous.4open.science/r/PITA_1/Table10.pdf, we first removed all terms involving first-order and second-order derivatives (marked as “None” in the table), which means that both the convective and diffusion terms are eliminated. Next, we removed only those library elements that include second-order derivatives (labeled “One-Order” in the table), thereby naturally eliminating the diffusion term while retaining the convective term. And the full library results are marked as “Full”. **It is evident that the full library achieves state-of-the-art predictive accuracy. In the "One-Order" configuration, where only the convective term is retained, the performance experiences a slight degradation. Further removal of both the convective and diffusion terms leads to a modest compromise in the model's ability to capture the underlying dynamics. Nevertheless, the model still achieves a 41.42% improvement over the baseline, demonstrating the robustness of the approach even with limited incorporation of physical knowledge.** This is consistent with our ablation results on the physics loss (see Section 4.3 Task 1), which show that even when the physics loss becomes ineffective, PITA still brings improvements over the baseline by relying on data loss and consistency loss.
> > >
> > > Furthermore, we directly introduced a gradient attack by injecting harmful salt-and-pepper noise into the gradient of the physical loss, which severely disrupts the learning of the underlying physical laws. As shown in https://anonymous.4open.science/r/PITA_1/Table9.pdf, under a high-intensity attack of 0.05, PITA loses its capability to capture the high-precision physics, yet still maintains baseline-level performance. This means that, in the worst-case scenario (which is highly unlikely), PITA would degrade to an auto-regressive model without causing catastrophic performance deterioration.
> > >
> > > [1] Gao Q, Zou M Y. An analytical solution for two and three-dimensional nonlinear Burgers' equation[J]. Applied Mathematical Modelling, 2017, 45: 255-270.
> > >
> > > Finally, we sincerely thank you for your valuable suggestions and **commit to implementing all the suggested revisions**. Should you find it appropriate, we would be grateful if you could consider **updating 'Overall Recommendation'** in the system to reflect your acceptance decision to the Area Chair.

---

### Official Review · Reviewer_SLyt · 2025-03-13

**Overall Recommendation:** 3

**Summary:**

The paper introduces Physics-informed Temporal Alignment (PITA), a new self-supervised learning framework aimed at improving autoregressive PDE foundation models. The authors identify a "shortcut" issue common in autoregressive models, where the model takes easy solutions, leading to accumulated prediction errors, especially in long trajectories or out-of-distribution data. To fix this issue, PITA combines physics-informed constraints with traditional autoregressive predictions. This is done by first discovering PDE equations from observation data and then aligning the predicted trajectories with these discovered physics constraints.

The main contributions are: (1) integrating PDE discovery with autoregressive prediction to solve the "shortcut" problem; (2) proposing a combined loss function with data, physics, and consistency terms, balanced using uncertainty-based weighting; and (3) demonstrating strong results across several PDE benchmarks (e.g., PDEBench, PDEArena, CFDBench), achieving better accuracy than existing foundation models, especially for long-term predictions.

**Claims And Evidence:**

Yes

**Essential References Not Discussed:**

No

**Experimental Designs Or Analyses:**

The comparisons with existing autoregressive PDE foundation models, such as DPOT and MPP, provide a reasonable basis for assessing the effectiveness of PITA. The use of ablation studies to analyze the impact of different components, including loss functions, downsampling strategies, and model sizes, strengthens the validity of the experimental results. While PITA improves accuracy, the additional cost of PDE discovery and alignment is not fully analyzed in terms of runtime or scalability to larger datasets and longer trajectories.

**Methods And Evaluation Criteria:**

Yes, it uses standard PDE datasets for evaluation, such as PDEBench, PDEArena, CFDBench.

**Other Comments Or Suggestions:**

N/A

**Other Strengths And Weaknesses:**

Strengths: (1) The idea of combining PDE discovery with autoregressive modeling is innovative, addressing the common shortcut issue clearly and effectively. (2) The paper is well-motivated, providing good insight into why existing autoregressive approaches might fail on long-term trajectories.

Limitations: (1) Although the approach significantly improves performance, it could be computationally intensive due to the additional complexity introduced by PDE discovery and alignment steps, which might limit its applicability in practice. (2) The sensitivity of the method to noisy or incomplete observational data isn't fully addressed; this aspect might substantially impact its applicability in real-world scenarios.

**Questions For Authors:**

1. How sensitive is the PDE discovery step in PITA to noise or incomplete observational data?
2. Why did you choose the sparse regression approach specifically for discovering PDE terms? Would other methods perform similarly or better?
3. How would PITA perform if PDE discovery fails to find the correct governing equations, especially when dealing with noisy or limited data?

**Relation To Broader Scientific Literature:**

The paper proposes physics-informed temporal alignment (PITA), a self-supervised learning framework to improve the accuracy of existing PDE foundation models.

**Theoretical Claims:**

The paper does not provide theoretical analysis or proofs; it is primarily empirical

---

> ### Author Rebuttal · Authors · 2025-04-01
>
> 1.Regarding the additional cost of PDE discovery and alignment.
>
> Reply 1: We appreciate the reviewer highlighting computational scaling. For fixed batch sizes, the additional time from PDE discovery and alignment remains constant, independent of dataset scale or trajectory length, since it depends solely on parameter $T_{ar}$ (Sec. 3.2). Furthermore, for single-dataset fine-tuning, pre-training incurs baseline cost only, while PITA overhead occurs exclusively during fine-tuning, making its overall training time negligible.
>
> 2.Regarding additional computational cost and its limitations.
>
> Reply 2: Thanks for your valuable suggestions. As shown in https://anonymous.4open.science/r/PITA_1/Table1.pdf, the time consumption of the PDE discovery and alignment steps is constant and does not scale with the model size. Moreover, PITA performs PDE discovery and alignment only during the training phase, so its inference efficiency is identical to the baseline model. Due to space limitations, please refer to the discussion in Reply 3 to Reviewer y7j5.
>
> 3.Regarding noise or incomplete observational data.
>
> Reply 3: We appreciate your concern about noise sensitivity. The L0 regularization (Sparse Regression) in our PDE discovery step inherently suppresses noise by enforcing sparsity in the candidate coefficient space. As noted in compressed sensing theory [5], sparse regularization, particularly L0 regularization, prunes small-magnitude terms caused by observational noise, effectively driving them to zero while preserving dominant dynamical terms [1, 2]. Additionally, prior work [3, 4] has provided experimental evidence that sparse regression can successfully identify governing equations with high accuracy even in the presence of noisy data. Regarding incomplete data, PITA employs local sampling in both time and space. To further demonstrate the robustness of PITA, we conducted experiments under two challenging settings: (1) noisy data with noise levels of 0.05, 0.005, 0.0005, and (2) incomplete data where 25% of the observations were randomly masked. As shown in the results https://anonymous.4open.science/r/PITA_1/Table6.pdf and Table7.pdf, PITA consistently outperforms baselines even under these adverse conditions. This illustrated the robustness of L0 regularization against noisy or incomplete observational data.
>
> 4.Regarding the choice of the sparse regression approach.
>
> Reply 4: As previously discussed in terms of noise robustness, sparse regression achieves the separation of physical law extraction from noise interference through explicit sparsity constraints. The L0-norm regularization adopted in this paper systematically eliminates noise-dominated low-magnitude terms through a hard thresholding mechanism. Whereas LASSO’s L1 regularization suffers from collinear feature bias due to convex relaxation [1], and STLS lacks inherent regularization in its recursive ordinary least squares framework, our method achieves principled sparsity control. Notably, while ridge regression’s L2 penalty improves matrix conditioning [3], it fundamentally conflicts with sparsity requirements through non-zero coefficient retention. This structured sparsity enforcement proves essential for disentangling physical laws from noise interference. Besides, in our experiments https://anonymous.4open.science/r/PITA_1/Table5.pdf, when employing other non-sparse methods such as pseudo-inverse or least squares, the resulting coefficient matrices exhibit significant non-physical oscillations due to the lack of sparsity constraints in the solution space. Such ill-conditioned solutions make it difficult to effectively normalize $L_{Phy}$ and can also trigger gradient explosion phenomena during backpropagation.
>
> 5.Regarding the failure cases of PDE discovery
>
> Reply 5: The discussion regarding noise and incomplete data has already been provided in Reply 3. In this discussion, we designed a specific experiment to evaluate PITA's robustness against dynamic failure, where we deliberately injected noise into the gradient computation of PITA's physics loss to simulate inaccurate or corrupted dynamics. As shown in https://anonymous.4open.science/r/PITA_1/Table9.pdf, even under such a direct attack on the physics constraint, PITA still exhibits strong robustness. We further validate robustness by perturbing the PDE spatial grid (see https://anonymous.4open.science/r/PITA_1/Table8.pdf), showing that even with an intentionally distorted grid, the method maintains performance parity with unperturbed settings. As discussed with Reply 3, even when the PDE discovery fails, the multi-task learning framework mitigates its negative impact, ensuring that the model degrades gracefully to the baseline performance.
>
> [1] Sparse representation for signal classification.
>
> [2] Robust sparse linear discriminant analysis.
>
> [3] Physics-informed learning of governing equations from scarce data.
>
> [4] Data-driven discovery of partial differential equations.
>
> [5] Compressed sensing.

---

### Decision · Program_Chairs · 2025-05-01

**Decision:**

Accept (poster)

**Comment:**

The paper introduces a self-supervised framework to mitigate error accumulation in auto-regressive PDE surrogates. The core contribution lies in integrating losses derived from governing equations with the standard data loss used for training surrogates. These governing equations are discovered from both the data and the model predictions using symbolic regression. Two loss terms are introduced: a physical loss corresponding to the inferred equation, and a so-called consistency loss that measures the similarity between the equation inferred from historical data and the one inferred from the model’s predictions. This framework is compatible with existing models, and experiments are conducted on various foundation model architectures. Results on 12 datasets demonstrate the positive impact of this approach on reducing error accumulation.

Reviewers requested additional experiments and clarifications regarding the added complexity of the method relative to the backbone models it builds upon, as well as its robustness to observational noise and to errors or incompleteness in the discovered equations. The authors provided detailed responses and supplementary experiments addressing these concerns. The reviewers’ feedback on the responses was generally positive: two out of four increased their scores, and all expressed an inclination toward acceptance.
I recommend that the authors carefully incorporate the reviewers’ suggestions and revise the manuscript, particularly to improve clarity in some imprecise sections. I recommend acceptance.